# Defending Against Adversarial Examples by Regularized Deep Embedding

## Abstract

Recent studies have demonstrated the vulnerability of deep convolutional neural networks against adversarial examples. Inspired by the observation that the intrinsic dimension of image data is much smaller than its pixel space dimension and the vulnerability of neural networks grows with the input dimension, we propose to embed high-dimensional input images into a low-dimensional space to perform classification. However, arbitrarily projecting the input images to a low-dimensional space without regularization will not improve the robustness of deep neural networks. We propose a new framework, Embedding Regularized Classifier (ER-Classifier), which improves the adversarial robustness of the classifier through embedding regularization. Experimental results on several benchmark datasets show that, our proposed framework achieves state-of-the-art performance against strong adversarial attack methods.

## 1 Introduction

Deep neural networks (DNNs) have been widely used for tackling numerous machine learning problems that were once believed to be challenging. With their remarkable ability of fitting training data, DNNs have achieved revolutionary successes in many fields such as computer vision, natural language progressing, and robotics. However, they were shown to be vulnerable to adversarial examples that are generated by adding carefully crafted perturbations to original images. The adversarial perturbations can arbitrarily change the network's prediction but often too small to affect human recognition (Szegedy et al., 2013; Kurakin et al., 2016). This phenomenon brings out security concerns for practical applications of deep learning.

Two main types of attack settings have been considered in recent research (Goodfellow et al.; Carlini & Wagner, 2017a; Chen et al., 2017; Papernot et al., 2017): black-box and white-box settings. In the black-box setting, the attacker can provide any inputs and receive the corresponding predictions. However, the attacker cannot get access to the gradients or model parameters under this setting; whereas in the white-box setting, the attacker is allowed to analytically compute the model's gradients, and have full access to the model architecture and weights. In this paper, we focus on defending against the white-box attack which is the harder task.

Recent work (Simon-Gabriel et al., 2018) presented both theoretical arguments and an empirical one-to-one relationship between input dimension and adversarial vulnerability, showing that the vulnerability of neural networks grows with the input dimension. Therefore, reducing the data dimension may help improve the robustness of neural networks. Furthermore, a consensus in the high-dimensional data analysis community is that, a method working well on the high-dimensional data is because the data is not really of high-dimension (Levina & Bickel, 2005). These high-dimensional data, such as images, are actually embedded in a low dimensional space. Hence, carefully reducing the input dimension may improve the robustness of the model without sacrificing performance.

Inspired by the observation that the intrinsic dimension of image data is actually much smaller than its pixel space dimension (Levina & Bickel, 2005) and the vulnerability of a model grows with its input dimension (Simon-Gabriel et al., 2018), we propose a defense framework that embeds input images into a low-dimensional space using a deep encoder and performs classification based on the latent embedding with a classifier network. However, an arbitrary projection does not guarantee improving the robustness of the model, because there are a lot of mapping functions including non-robust ones from the raw input space to the low-dimensional space capable of minimizing the classification loss. To constrain the mapping function, we employ distribution regularization in the

embedding space leveraging optimal transport theory. We call our new classification framework Embedding Regularized Classifier (ER-Classifier). To be more specific, we introduce a discriminator in the latent space which tries to separate the generated code vectors from the encoder network and the ideal code vectors sampled from a prior distribution, i.e., a standard Gaussian distribution. Employing a similar powerful competitive mechanism as demonstrated by Generative Adversarial Networks (Goodfellow et al., 2014), the discriminator enforces the embedding space of the model to follow the prior distribution.

In our ER-Classifier framework, the encoder and discriminator structures together project the input data to a low-dimensional space with a nice shape, then the classifier performs prediction based on the low-dimensional embedding. Based on the optimal transport theory, the proposed ER-Classifier minimizes the discrepancy between the distribution of the true label and the distribution of the framework output, thus only retaining important features for classification in the embedding space. With a small embedding dimension, the effect of the adversarial perturbation is largely diminished through the projection process.

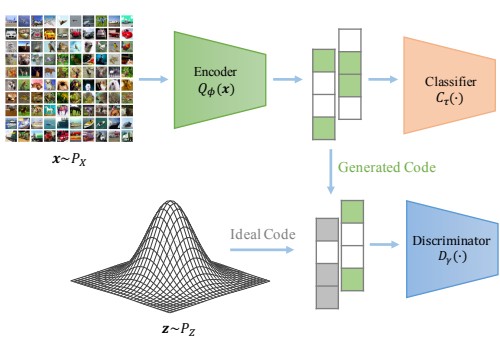

Figure 1: Overview of ER-Classifier framework

We compare ER-Classifier with other state-of-the-art defense methods on MNIST, CIFAR10, STL10 and Tiny Imagenet. Experimental results demonstrate that our proposed ER-Classifier outperforms other methods by a large margin. To sum up, this paper makes the following three main contributions:

- A novel unified end-to-end robust deep neural network framework against adversarial attacks is proposed, where the input image is first projected to a low-dimensional space and then classified.
- An objective is induced to minimize the optimal transport cost between the true class distribution and the framework output distribution, guiding the encoder and discriminator to project the input image to a low-dimensional space without losing important features for classification.
- Extensive experiments demonstrate the robustness of our proposed ER-Classifier framework under the white-box attacks, and show that ER-Classifier outperforms other state-of-the-art approaches on several benchmark image datasets.

As far as we know, our approach is the first that applies optimal transport theory, i.e., a Wasserstein distance regularization, to a bottleneck embedding layer of a deep neural network in a purely supervised learning setting without considering any reconstruction loss, although optimal transport theory or a discriminator loss has been applied to generative models in an unsupervised learning setting (Makhzani et al., 2015; Tolstikhin et al., 2017); (2) Our method is also the first that establishes the connection between a Wasserstein distance regularization and the robustness of deep neural networks for defending against adversarial examples.

## 2 RELATED WORK

In this section, we summarize related work into three categories: attack methods, defense mechanisms and optimal transport theory. We first discuss different white-box attack methods, followed by a description of different defense mechanisms against, and finally optimal transport theory.

### 2.1 ATTACK METHODS

Under the white-box setting, attackers have all information about the targeted neural network, including network structure and gradients. Most white-box attacks generate adversarial examples based on the gradient of loss function with respect to the input. An algorithm called fast gradient sign method (FGSM) was proposed in (Goodfellow et al.) which generates adversarial examples based on the sign of gradient. Many other white-box attack methods have been proposed

recently (Moosavi-Dezfooli et al., 2016; Chen et al., 2018; Madry et al., 2017; Carlini & Wagner, 2017b), and among them C&W and PGD attacks have been widely used to test the robustness of machine learning models.

**C&W attack:** The adversarial attack method proposed by Carlini and Wagner (Carlini & Wagner, 2017b) is one of the strongest white-box attack methods. They formulate the adversarial example generating process as an optimization problem. The proposed objective function aims at increasing the probability of the target class and minimizing the distance between the adversarial example and the original input image. Therefore, C&W attack can be viewed as a gradient-descent based adversarial attack.

**PGD attack:** The projected gradient descent attack is proposed by (Madry et al., 2017), which finds adversarial examples in an $\epsilon$-ball of the image. The PGD attack updates in the direction that decreases the probability of the original class most, then projects the result back to the $\epsilon$-ball of the input. An advantage of PGD attack over C&W attack is that it allows direct control of distortion level by changing $\epsilon$, while for C&W attack, one can only do so indirectly via hyper-parameter tuning.

Both C&W attack and PGD attack have been frequently used to benchmark the defense algorithms due to their effectiveness (Athalye et al., 2018). In this paper, we mainly use $l_\infty$-PGD untargeted attack to evaluate the effectiveness of the defense method under white-box setting.

Instead of crafting different adversarial perturbation for different input image, an algorithm was proposed by (Moosavi-Dezfooli et al., 2017) to construct a universal perturbation that causes natural images to be misclassified. However, since this universal perturbation is image-agnostic, it is usually larger than the image-specific perturbation generated by PGD and C&W.

## 2.2 DEFENSE MECHANISMS

Many works have been done to improve the robustness of deep neural networks. To defend against adversarial examples, defenses that aim to increase model robustness fall into three main categories: i) augmenting the training data with adversarial examples to enhance the existing classifiers (Madry et al., 2017; Na et al., 2017; Goodfellow et al.); ii) leveraging model-specific strategies to enforce model properties such as smoothness (Papernot et al., 2016); and, iii) trying to remove adversarial perturbations from the inputs (Xie et al., 2017; Samangouei et al., 2018; Meng & Chen, 2017). We select three representative methods that are effective under white-box setting.

**Adversarial training:** Augmenting the training data with adversarial examples can increase the robustness of the deep neural network. Madry et al. (Madry et al., 2017) recently introduced a min-max formulation against adversarial attacks. The proposed model is not only trained on the original dataset but also adversarial example in the $\epsilon$-ball of each input image.

**Random Self-Ensemble:** Another effective defense method under white-box setting is RSE (Liu et al., 2017). The authors proposed a "noise layer", which fuses output of each layer with Gaussian noise. They empirically show that the noise layer can help improve the robustness of deep neural networks. The noise layer is applied in both training and testing phases, so the prediction accuracy will not be largely affected.

**Defense-GAN:** Defense-GAN (Samangouei et al., 2018) leverages the expressive capability of GANs to defend deep neural networks against adversarial examples. It is trained to project input images onto the range of the GAN's generator to remove the effect of the adversarial perturbation. Another defense method that uses the generative model to filter out noise is MagNet proposed by (Meng & Chen, 2017). However, the differences between ER-Classifier and the two methods are obvious. ER-Classifier focuses on reducing the dimension, and performing classification based on the low-dimensional embedding, while Defense-GAN and MagNet mainly apply the generative model to filter out the adversarial noise, and both Defense-GAN and MagNet perform classification on the original dimension space. (Samangouei et al., 2018) showed that Defense-GAN is more robust than MagNet, so we only compare with Defense-GAN in the experiments.

**Other Related Methods:** Zhang et al. (2018) regularizes the latent space with Gaussian Mixture Model and applies KL-divergence to do the optimization. However, our method employs a simple but nice-shaped Gaussian prior for Wasserstein distance minimization to constrain the global shape of the latent embeddings, while permitting high freedom for the shapes of individual class distributions of latent embeddings. We want the classifier to decide the optimal class-specific distributions of

latent embeddings. Miyato et al. (2018) shares a similar idea to adversarial learning, but it employs virtual labels generated by a current classifier to identify search directions that can smooth the output label distribution of the classifier and is best suitable for semi-supervised learning. Please note that both methods in (Zhang et al., 2018; Miyato et al., 2018) are designed for improving generalization performance but not for defending against adversarial examples. A recent paper (Ilyas et al., 2019) shows that adversarial examples are purely human phenomenon and models tend to learn features that are not robust yet generalize well. We show that our Wasserstein distance regularization helps to identify robust features, which will be discussed later.

**Notations**   In this paper, we use $l_\infty$ and $l_2$ distortion metrics to measure similarity. We report $l_\infty$ distance in the normalized $[0, 1]$ space, so that a distortion of $0.031$ corresponds to $8/256$, and $l_2$ distance as the total root-mean-square distortion normalized by the total number of pixels.

We use calligraphic letters for sets (i.e., $\mathcal{X}$), capital letters for random variables (i.e., $X$), and lower case letters for their values (i.e., $x$). The probability distributions are denoted with capital letters (i.e., $P_X$) and corresponding densities with lower case letters (i.e., $p_X$).

## 3   PROPOSED FRAMEWORK: EMBEDDING REGULARIZED CLASSIFIER

### 3.1   FRAMEWORK DETAILS

We propose a novel defense framework, ER-Classifier, which aims at projecting the image data to a low-dimensional space to remove noise and stabilize the classification model by minimizing the optimal transport cost between the true label distribution $P_Y$ and the distribution of the ER-Classifier output ($P_C$). An overview of the framework is shown in Figure 1. The encoder and discriminator structures together help diminish the effect of the adversarial perturbation by projecting input data to a space of lower dimension, then the classifier part performs classification based on the low-dimensional embedding.

Mathematically, input images $X \in \mathcal{X} = \mathbb{R}^d$ are projected to a low-dimensional embedding vector $Z \in \mathcal{Z} = \mathbb{R}^k$ through the encoder $\boldsymbol{Q}_\phi$. The discriminator $\boldsymbol{D}_\gamma$ discriminates between the generated code $\tilde{Z} \sim \boldsymbol{Q}_\phi(Z|X)$ and the ideal code $Z \sim P_Z$. The classifier $\boldsymbol{C}_\tau$ performs classification based on the generated code $\tilde{Z}$, producing output $U \in \mathcal{U} = \mathbb{R}^m$, where $m$ is the number of classes. The label of $X$ is denoted as $Y \in \mathcal{U}$.

The Kantorovich's distance induced by the optimal transport problem is given by

$$W_c(P_Y, P_C) \coloneqq \inf_{\Gamma \in \mathcal{P}(Y \sim P_Y, U \sim P_C)} \mathbb{E}_{(Y,U) \sim \Gamma} \left\{ c(Y, U) \right\},$$

where $\mathcal{P}(Y \sim P_Y, U \sim P_C)$ is the set of all joint distributions of $(Y, U)$ with marginals $P_Y$ and $P_C$, and $c(y, u) : \mathcal{U} \times \mathcal{U} \mapsto \mathbb{R}_+$ is any measurable cost function. $W_c(P_Y, P_C)$ measures the divergence between probability distributions $P_Y$ and $P_C$. When the probability measures are on a metric space, the $p$-th root of $W_c$ is called the $p$-Wasserstein distance.

To minimize the Wasserstein distance between the distribution of the true label ($P_Y$) and the distribution of the ER-Classifier output ($P_C$), we can prove that it is sufficient to find a conditional distribution $\boldsymbol{Q}(Z|X)$ such that its marginal distribution $\boldsymbol{Q}_Z$ is identical to a prior distribution $P_Z$. The theorem and the proof are deferred to the Appendix. In this paper, we apply standard Gaussian as our prior distribution $P_Z$, but other priors may be used for different cases. The final objective of ER-Classifier is:

$$\inf_{\boldsymbol{Q}(Z|X) \in \mathcal{Q}} \mathbb{E}_{P_X} \mathbb{E}_{\boldsymbol{Q}(Z|X)} \left\{ \ell(f(X), \boldsymbol{C}(Z)) \right\} + \lambda \mathcal{D}(\boldsymbol{Q}_Z, P_Z), \tag{1}$$

where $\mathcal{Q}$ can be a deterministic encoder as focused by this paper due to its simplicity or stochastic encoder as the one in a standard Variational Autoencoder, $\lambda > 0$ is a hyper-parameter and $\mathcal{D}$ is an arbitrary divergence between $\boldsymbol{Q}_Z$ and $P_Z$.

To estimate the divergences between $\boldsymbol{Q}_Z$ and $P_Z$, we apply a GAN-based framework, fitting a discriminator to minimize the 1-Wasserstein distance between $\boldsymbol{Q}_Z$ and $P_Z$:

$$W(\boldsymbol{Q}_Z, P_Z) = \inf_{\Gamma \in \mathcal{P}(\tilde{Z} \sim \boldsymbol{Q}_Z, Z \sim P_Z)} \mathbb{E}_{(\tilde{Z}, Z) \sim \Gamma} \| \tilde{Z} - Z \|.$$

We have also tried the Jensen-Shannon divergence, but as expected, Wasserstein distance provides more stable training and better results. When training the framework, the weight clipping method proposed in Wasserstein GAN (Arjovsky et al., 2017) is applied to help stabilize the training of discriminator $D_\gamma$. The training algorithm is summarized in Algorithm 1.

At training stage, the encoder $Q_\phi$ first maps the input $x$ to a low-dimensional space, resulting in generated code ($\tilde{z}$). Another ideal code ($z$) is sampled from the prior distribution, and the discriminator $D_\gamma$ discriminates between the ideal code (positive data) and the generated code (negative data). The classifier ($C_\tau$) predicts the image label based on the generated code ($\tilde{z}$).

---

**Algorithm 1** Training ER-Classifier

1: **Input:** Regularization coefficient $\lambda > 0$, encoder $Q_\phi$, discriminator $D_\gamma$, and classifier $C_\tau$.
2: **Note:** $\ell$ stands for the cross-entropy loss.
3: **while** $(\phi, \gamma, \tau)$ not converged **do**
4:     Sample $\{(x_1, y_1), ..., (x_n, y_n)\}$ from the training set
5:     Sample $\{z_1, ..., z_n\}$ from the prior $P_Z$
6:     Directly obtain or sample $\tilde{z}_i$ from $Q_\phi(Z|x_i)$ for $i = 1, ..., n$
7:     Update $D_\gamma$ by ascending the following objective by 1-step Adam:

$$\frac{\lambda}{n} \sum_{i=1}^{n} D_\gamma(z_i) - D_\gamma(\tilde{z}_i)$$

8:     Update $Q_\phi$ and $C_\tau$ by descending the following objective by 1-step Adam:

$$\frac{1}{n} \sum_{i=1}^{n} \ell(C_\tau(Q_\phi(x_i)), y_i)$$

9:     Update $Q_\phi$ by ascending the following objective by 1-step Adam:

$$\frac{\lambda}{n} \sum_{i=1}^{n} D_\gamma(Q_\phi(x_i))$$

10: **end while**

---

At inference time, only the encoder $Q_\phi$ and the classifier $C_\tau$ are used. The input image $x$ is first mapped to a low-dimensional space by the encoder ($\tilde{z} = Q_\phi(x)$), then the latent code $\tilde{z}$ is fed into the classifier to obtain the predicted label.

The main goal of ER-Classifier is leveraging input space dimension reduction to remove adversarial perturbations. Therefore, other defense methods can also benefit from this property. Our framework is trained with min-max robust optimization (Madry et al., 2017).

## 3.2 JUSTIFICATIONS OF OUR FRAMEWORK

There are two Wasserstein distances (W-distances) in our framework. One is the W-distance between the aggregated latent embedding distribution $Q(z)$ and the prior distribution $P_Z$, and the other one is the W-distance between the true label distribution $P_Y$ and the distribution of the ER-Classifier output ($P_C$). In Algorithm 1, we are minimizing the first one. The theorem in the Appendix shows that minimizing the first W-distance in combination with minimizing a standard cross-entropy loss as done in Algorithm 1 is equivalent to minimizing the second W-distance, which guarantees that the training process is not distracted from the main goal of the framework, classification. That is to say, Algorithm 1 will result in a classifier with the following property: the global output distribution of the classifier will match the global ground-truth label distribution in the data no matter whether the encoder $Q_\phi$ is deterministic or stochastic (the second W-distance is automatically minimized).

It's hard to analyze the importance of the theorem in the Appendix if we just look at a deterministic encoder. Let's convert this deterministic encoder to a stochastic encoder that outputs a Gaussian $z$ with a fixed variance $\epsilon$ and the mean being the same as its corresponding deterministic version. The theory tells us that, by minimizing the first W-distance over all sampled $z$'s from this stochastic encoder and the standard cross-entropy loss, we will automatically minimize the second W-distance

and preserve the global label frequency in the dataset, even though these $z$'s are only $\epsilon$-close to the deterministic encoding features of training data.

Moreover, we find that minimizing the W-distance helps the encoder identify some robust features instead of non-robust features (Ilyas et al., 2017), because our proposed regularization constrains the $\epsilon$-ball around each $\boldsymbol{Q}_\phi(Z|X)$ to contribute to preserving the global label distribution in the data, even with $X$ integrated out. From this perspective, we can view our proposed framework as "a supervised variant" of Generative Adversarial Network or Wasserstein Autoencoder in which the Generator or Decoder is replaced by a Classifier that generates labels from low-dimensional latent embeddings preserving global label frequency in the training dataset. Replacing W-distance with KL divergence loses all these nice properties.

In our framework, we use a simple but nice-shaped Gaussian prior $P_Z$ for W-distance minimization to constrain the global shape of the latent embeddings, while permitting high freedom for the shapes of individual class distributions of latent embeddings. We want the classifier to decide the optimal class-specific distributions of latent embeddings. In addition, it is interesting to explore how to set $\epsilon$-ball to make sure the stochastic encoder to best align the latent embedding $z$ to human-perceived robust features, which will be left as future work.

## 4 EXPERIMENTS

In this section, we compare the performance of our proposed algorithm (ER-Classifier) with other state-of-the-art defense methods on several benchmark datasets:

- MNIST (LeCun, 1998): handwritten digit dataset, which consists of $60,000$ training images and $10,000$ testing images. Theses are $28 \times 28$ black and white images in ten different classes.
- CIFAR10 (Krizhevsky & Hinton, 2009): natural image dataset, which contains $50,000$ training images and $10,000$ testing images in ten different classes. These are low resolution $32 \times 32$ color images.
- STL10 (Coates et al., 2011): color image dataset similar to CIFAR10, but contains only $5,000$ training images and $8,000$ testing images in ten different classes. The images are of higher resolution $96 \times 96$.
- Tiny Imagenet (Deng et al., 2009): a subset of Imagenet dataset. Tiny Imagenet has 200 classes, and each class has $500$ training images, $50$ testing images, making it a challenging benchmark for defense task. The resolution of the images is $64 \times 64$.

Various defense methods have been proposed to improve the robustness of deep neural networks. Here we compare our algorithm with state-of-the-art methods that are robust in white-box setting. Madry's adversarial training (**Madry's Adv**) has been recognized as one of the most successful defense method in white-box setting, as shown in (Athalye et al., 2018).

Random Self-Ensemble (**RSE**) method introduced by (Liu et al., 2017) adds stochastic components in the neural network, achieving similar performance to Madry's adversarial training algorithm.

Another method we would like to compare with is **Defense-GAN** (Samangouei et al., 2018). It first trains a generative adversarial network to model the distribution of the training data. At inference time, it finds a close output to the input image and feed that output into the classifier. This process "projects" input images onto the range of GAN's generator, which helps remove the effect of adversarial perturbations. In (Samangouei et al., 2018), the author demonstrated the performance of Defense-GAN on MNIST and Fashion-MNIST, so we will compare our method with Defense-GAN on MNIST.

Since the main goal of ER-Classifier is using dimension reduction to improve adversarial robustness, other defense methods can also benefit from this property. The proposed **ER-Classifier** is trained with min-max robust optimization (Madry et al., 2017). To demonstrate the dimension reduction ability of ER-Classifier, we include a variant **ER-Classifier$^-$** which trains ER-Classifier without min-max robust optimization.

### 4.1 EVALUATE MODELS UNDER WHITE-BOX $l_\infty$-PGD ATTACK

In this section, we evaluate the defense methods against $l_\infty$-PGD untargeted attack, which is one of the strongest white-box attack methods. Models are evaluated under different distortion level ($\epsilon$),

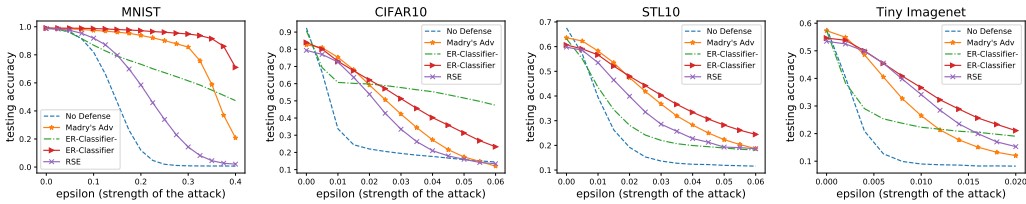

Figure 2: Testing accuracy under $l_\infty$-PGD attack on four different datasets: MNIST, CIFAR10, STL10 and Tiny Imagenet.

and the larger the distortion the stronger the attack. Depending on the image scale and type, different datasets are sensitive to different strength of attack.

Models on MNIST are evaluated under distortion level from $0$ to $0.4$ by $0.025$. Models on CIFAR10 and STL10 are evaluated under $\epsilon \in [0, 0.06, 0.005]$. Models on Tiny Imagenet are evaluated under $\epsilon \in [0, 0.02, 0.002]$. As mentioned in the notation part, all the distortion levels are reported in the normalized $[0, 1]$ space. The experimental results are shown in Figure 2. To demonstrate the results more clearly, we show part of the results in Table 1.

Based on Figure 2 and Table 1, we can see that ER-Classifier is the most robust one on a variety of datasets. ER-Classifier without min-max robust optimization can also improve the robustness of deep neural network. Compare the performance of ER-Classifier$^-$ with the performance of model without defense method (No Defense), we can see that ER-Classifier$^-$ is much more robust than the model with no defense method on all benchmark datasets. Besides, when the distortion level ($\epsilon$) is large, ER-Classifier$^-$ tends to perform better than other state-of-the-art defense methods on MNIST, CIFAR10 and Tiny Imagenet. This phenomenon is obvious on CIFAR10 and it even performs better than ER-Classifier when the attack strength is strong. The reason might be that without min-max robust optimization, it is easier to regularize the embedding space.

| Data | Defense | 0 | 0.1 | 0.2 | 0.3 | 0.4 |
|---|---|---|---|---|---|---|
| MNIST | Madry's Adv | 98.7 | 97.5 | 93.8 | 85.5 | 20.8 |
| | ER-Classifier | **99.1** | **98.7** | **97.2** | **94.9** | **71.1** |

| Data | Defense | 0 | 0.015 | 0.03 | 0.045 | 0.06 |
|---|---|---|---|---|---|---|
| CIFAR10 | Madry's Adv | 82.6 | **68.0** | 42.3 | 21.6 | 12.0 |
| | ER-Classifier | **84.0** | 67.5 | **51.3** | **35.8** | **23.3** |
| STL10 | Madry's Adv | **63.6** | 53.5 | 36.8 | 25.0 | 18.7 |
| | ER-Classifier | 60.7 | **52.1** | **40.3** | **30.6** | **24.5** |

| Data | Defense | 0 | 0.004 | 0.01 | 0.016 | 0.02 |
|---|---|---|---|---|---|---|
| Tiny Imagenet | Madry's Adv | **57.3** | 48.6 | 26.5 | 15.1 | 12.0 |
| | ER-Classifier | 54.6 | **50.0** | **36.7** | **25.6** | **21.1** |

Table 1: Testing accuracy (%) under different strength of PGD attacks. The table shows the results of ER-Classifier and Madry's adversarial training (Madry's Adv). The better accuracy is marked in **bold**.

We also compare Defense-GAN with our method ER-Classifier on MNIST. Although Defense-GAN was shown to be partly broken by (Athalye et al., 2018; Ilyas et al., 2017), both ER-Classifier and Defense-GAN share the similar idea of projecting the input to a learned manifold, and comparing to Defense-GAN is important to demonstrate the advantage of our novel Wassserstein distance regularization. Please note that Defense-GAN is not our major comparison baseline in

| Method | Testing Accuracy |
|---|---|
| Defense-GAN | 55.0 |
| ER-Classifier | 99.1 |

Table 2: Testing accuracy (%) of two defense methods under C&W attack with $l_2 \leq 0.005$.

this paper. Both ER-Classifier and Defense-GAN are evaluated against the $l_2$-C&W untargeted attack, one of the strongest white-box attack proposed in (Carlini & Wagner, 2017b). Defense-GAN is evaluated using the method proposed in (Athalye et al., 2018), and the code is available on github[1]. ER-Classifier is evaluated against $l_2$-C&W untargeted attack with the same hyper-parameter values as those used in the evaluation of Defense-GAN. The results under $l_2 \leq 0.005$ threshold are shown in Table 2. Based on Table 2, ER-Classifier is much more robust than Defense-GAN un-

---

[1]Publicly available at `https://github.com/anishathalye/obfuscated-gradients/tree/master/defensegan`

Figure 3: Testing accuracy of E-CLA, VAE-CLA and ER-Classsifier$^-$ under $l_\infty$-PGD attack on four different datasets: MNIST, CIFAR10, STL10 and Tiny Imagenet.

der the $l_2 \leq 0.005$ threshold. Since (Samangouei et al., 2018) did not evaluate Defense-GAN on CIFAR10, STL10 and Tiny Imagenet, without details of GAN structure, we can not compare with Defense-GAN on these datasets.

## 4.2 EVALUATE MODELS UNDER BLACK-BOX ATTACK

We evaluate Madry's adversarial training, ER-Classifier, and ER-Classifier$^-$, against a recently proposed black-box attack method called Nattack (Li et al., 2019)[2] on CIFAR10. Nattack is only performed on the first 100 images of CIFAR10 since the attack process takes a long time. We report the accuracy = number of correctly classified / number of attacked images (exactly 100). The accuracy of Madry's adv, ER-Classifier, and ER-Classifier$^-$ is, respectively, 38%, 43%, and 32%. ER-Classifier still outperforms Madry's adv.

## 4.3 EVALUATE THE EFFECT OF DISCRIMINATOR

ER-Classifier framework consists of three parts, and the classification task is done by the encoder $Q_\phi$ and classifier $C_\tau$. Without the discriminator part, the encoder can also project the input images to a low-dimensional space. However, arbitrarily projecting the images to a low-dimensional space with only the encoder part cannot improve the robustness of the model. In contrast, sometimes it even decreases the robustness of the model.

To show that arbitrarily projecting the input images to a low-dimensional space can not improve the robustness, we fit a framework with only the encoder and classifier part (E-CLA), where the encoder and classifier have the same structures as in ER-Classifier, and compare E-CLA with the ER-Classifier framework. For a fair comparison, both structures are trained without min-max robust optimization. The results are shown in Figure 3.

Based on Figure 3, we can observe that ER-Classifier is much more robust than just the encoder and classifier structure on MNIST, CIFAR10 and Tiny Imagenet. It is also more robust on STL10 but not that much. The reason might be that there are only $5,000$ training images in STL10 and the resolution is $96 \times 96$. Therefore, it is harder to learn a good embedding with limited amount of images. However, even when the number of training images is limited, ER-Classifier is still much more robust than the E-CLA structure. This observation demonstrates that regularization on the embedding space helps improve the adversarial robustness. Notice that the performance of E-CLA structure is similar to the performance of model without defense method on CIFAR10, STL10 and Tiny Imagenet, and worse on MNIST, which means the robustness of ER-Classifier does not come from the structure design.

Variational auto-encoder can project the images to low-dimensional space and use Kullback–Leibler divergence loss to regularize the embedding distribution, which does not need discriminator structure. Therefore, we also tried VAE-CLA, which applies Variational auto-encoder structure to do the projection and regularization. The experimental results in Figure 3 show that VAE-CLA does not perform as well as ER-Classifier. Based on the observation of the Kullback–Leibler loss and classification loss during the training process, it seems difficult for VAE-CLA to balance between the two tasks. The reason might be that Kullback–Leibler distances are not sensible cost functions when learning distributions supported by low dimensional manifolds (Arjovsky et al., 2017).

---

[2]https://github.com/Cold-Winter/Nattack

### 4.4 PRIOR SELECTION

ER-Classifier does not have restrictions on the choice of prior. However, the selection of prior is important as it imposes different restrictions on the embedding space.

Three different prior distributions are tested on MNIST and CIFAR10 datasets. They are standard Gaussian, Uniform$(-3, 3)$ and Cauchy$(0, 1)$, where Cauchy$(0, 1)$ has the same support as standard Gaussian but is heavy tailed and $99.7\%$ of the standard Gaussian points lies within $[-3, 3]$. All the models are trained without min-max robust optimization, and the experimental results are shown in Figure 4. Based on the results, all three priors work well, but standard Gaussian performs best on both datasets.

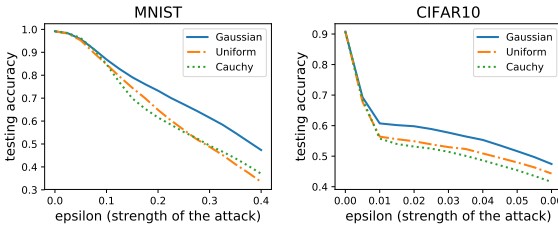

Figure 4: Testing accuracy of models with different prior distributions under $l_\infty$-PGD attack.

Ding et al. (Ding et al., 2019) prove that adversarial robustness is sensitive to the input data distribution, and if the data is uniformly distributed in the input space, no algorithm can achieve good robustness. They also empirically show that cornered/concentrated data distributions tend to achieve better robustness. This helps explain why regularizing the embedding space can help improve robustness. Though the projection process reduces the input dimension, the embedding space is still large. Prior distribution helps push the embedding space to be more concentrated, reducing the valid perturbation space.

Details of hyper-parameter selection, model structure and code are included in the supplementary part. Embedding space visualization can also be found in the supplementary material.

## 5 CONCLUSION

In this paper, we propose a new defense framework, ER-Classifier, which projects the input images to a low-dimensional space to remove adversarial perturbation and stabilize the model through minimizing the discrepancy between the true label distribution and the framework output distribution. We empirically show that ER-Classifier is much more robust than other state-of-the-art defense methods on several benchmark datasets. Future work will include further exploration of the low-dimensional space to improve the robustness of deep neural network.

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

APPENDIX

## THEORETICAL ANALYSIS OF OUR FRAMEWORK

Mathematically, input images $X \in \mathcal{X} = \mathbb{R}^d$ are projected to a low-dimensional embedding vector $Z \in \mathcal{Z} = \mathbb{R}^k$ through the encoder $\boldsymbol{Q}_\phi$. The discriminator $\boldsymbol{D}_\gamma$ discriminates between the generated code $\tilde{Z} \sim \boldsymbol{Q}_\phi(Z|X)$ and the ideal code $Z \sim P_Z$. The classifier $\boldsymbol{C}_\tau$ performs classification based on the generated code $\tilde{Z}$, producing output $U \in \mathcal{U} = \mathbb{R}^m$, where $m$ is the number of classes. The label of $X$ is denoted as $Y \in \mathcal{U}$.

The ER-Classifier framework embeds important classification features by minimizing the discrepancy between the distribution of the true label ($P_Y$) and the distribution of the framework output ($P_C$). In the framework, the classifier ($P_C(U|Z)$) maps a latent code $Z$ sampled from a fixed distribution in a latent space $\mathcal{Z}$, to the output $U \in \mathcal{U} = \mathbb{R}^m$. The density of ER-Classifier output is defined as follow:

$$p_C(u) := \int_{\mathcal{Z}} p_C(u|z) p_Z(z) dz, \ \ \forall u \in \mathcal{U}. \tag{2}$$

In this paper we apply standard Gaussian as our prior distribution $P_Z$, but other priors may be used for different cases. Assume there is an oracle $f : \mathcal{X} \mapsto \mathcal{U}$ assigning the image data ($X \in \mathcal{X}$) its true label ($Y \in \mathcal{U}$). We want to minimize the optimal transport cost between the distribution of the true label ($P_Y$) and the distribution of the ER-Classifier output ($P_C$).

### OPTIMAL TRANSPORT THEORY

There are various ways to define the distance or divergence between the target distribution and the model distribution. In this paper, we turn to the optimal transport theory (Villani, 2008), which provides a much weaker topology than many others. In real applications, data is usually embedded in a space of a much lower dimension, such as a non-linear manifold. *Kullback-Leibler* divergence, *Jensen-Shannon* divergence and *Total Variation* distance are not sensible cost functions when learning distributions supported by lower dimensional manifolds (Arjovsky et al., 2017). In contrast, the optimal transport cost is more sensible in this setting. Kantorovich's distance induced by the optimal transport problem is given by

$$W_c(P_Y, P_C) := \inf_{\Gamma \in \mathcal{P}(Y \sim P_Y, U \sim P_C)} \mathbb{E}_{(Y,U) \sim \Gamma} \left\{ c(Y, U) \right\},$$

where $\mathcal{P}(Y \sim P_Y, U \sim P_C)$ is the set of all joint distributions of $(Y, U)$ with marginals $P_Y$ and $P_C$, and $c(y, u) : \mathcal{U} \times \mathcal{U} \mapsto \mathbb{R}_+$ is any measurable cost function. $W_c(P_Y, P_C)$ measures the divergence between probability distributions $P_Y$ and $P_C$. When the probability measures are on a metric space, the $p$-th root of $W_c$ is called the $p$-Wasserstein distance.

To minimize the optimal transport cost between the distribution of the true label ($P_Y$) and the distribution of the ER-Classifier output ($P_C$), it is sufficient to find a conditional distribution $\boldsymbol{Q}(Z|X)$ such that its marginal distribution $\boldsymbol{Q}_Z$ is identical to the prior distribution $P_Z$.

**Theorem 1** *For $P_C$ as defined above with a deterministic $P_C(U|Z)$ and any function $\boldsymbol{C} : \mathcal{Z} \mapsto \mathcal{U}$*

$$\inf_{\Gamma \in \mathcal{P}(Y \sim P_Y, U \sim P_C)} \mathbb{E}_{(Y,U) \sim \Gamma} \left\{ \ell(Y, U) \right\}$$
$$= \inf_{\boldsymbol{Q}: \boldsymbol{Q}_Z = P_Z} \mathbb{E}_{P_X} \mathbb{E}_{\boldsymbol{Q}(Z|X)} \left\{ \ell(f(X), \boldsymbol{C}(Z)) \right\},$$

*where $\Gamma \in \mathcal{P}(Y \sim P_Y, U \sim P_C)$ is the set of all joint distributions of $(Y, U)$ with marginals $P_Y$ and $P_C$, and $\ell(y, u) : \mathcal{U} \times \mathcal{U} \mapsto \mathbb{R}_+$ is any measurable cost function. $\boldsymbol{Q}_Z$ is the marginal distribution of $Z$ when $X \sim P_X$ and $Z \sim \boldsymbol{Q}(Z|X)$. (The proof is presented later.)*

Therefore, optimizing over the objective on the r.h.s is equivalent to minimizing the discrepancy between the true label distribution ($P_Y$) and the output distribution $P_C$, thus the important classification features are embedded in the low-dimensional space. This is the core idea of the paper,

summarizing the high-dimensional data in a space of much lower dimension without losing important features for classification. To implement the r.h.s objective, the constraint on $\boldsymbol{Q}_Z$ can be relaxed by adding a penalty term. The final objective of ER-Classifier is:

$$\inf_{\boldsymbol{Q}(Z|X)\in\mathcal{Q}} \mathbb{E}_{P_X}\mathbb{E}_{\boldsymbol{Q}(Z|X)}\left\{\ell(f(X),\boldsymbol{C}(Z))\right\} + \lambda\mathcal{D}(\boldsymbol{Q}_Z,P_Z), \tag{3}$$

where $\mathcal{Q}$ is any nonparametric set of probabilistic encoders, $\lambda > 0$ is a hyper-parameter and $\mathcal{D}$ is an arbitrary divergence between $\boldsymbol{Q}_Z$ and $P_Z$.

To estimate the divergences between $\boldsymbol{Q}_Z$ and $P_Z$, we apply a GAN-based framework, fitting a discriminator to minimize the 1-Wasserstein distance between $\boldsymbol{Q}_Z$ and $P_Z$:

$$W(\boldsymbol{Q}_Z,P_Z) = \inf_{\Gamma\in\mathcal{P}(\tilde{Z}\sim\boldsymbol{Q}_Z,Z\sim P_Z)} \mathbb{E}_{(\tilde{Z},Z)\sim\Gamma}\|\tilde{Z}-Z\|.$$

We have also tried the Jensen-Shannon divergence, but as expected, Wasserstein distance provides more stable training and better results. When training the framework, the weight clipping method proposed in Wasserstein GAN (Arjovsky et al., 2017) is applied to help stabilize the training of discriminator $\boldsymbol{D}_\gamma$.

## PROOF OF THEOREM 1

The proof of Theorem 1 is adapted from the proof of Theorem 1 in (Tolstikhin et al., 2017). Consider certain sets of joint probability distributions of three random variables $(X,U,Z)\in\mathcal{X}\times\mathcal{U}\times\mathcal{Z}$. $X$ can be taken as the input images, $U$ as the output of the framework, and $Z$ as the latent codes. $P_{\boldsymbol{C},Z}(U,Z)$ represents a joint distribution of a variable pair $(U,Z)$, where $Z$ is first sampled from $P_Z$ and then $U$ from $P_{\boldsymbol{C}}(U|Z)$. $P_{\boldsymbol{C}}$ defined in (2) is the marginal distribution of $U$ when $(U,Z)\sim P_{\boldsymbol{C},Z}$.

The joint distributions $\Gamma(X,U)$ or couplings between values of $X$ and $U$ can be written as $\Gamma(X,U) = \Gamma(U|X)P_X(X)$ due to the marginal constraint. $\Gamma(U|X)$ can be decomposed into an encoding distribution $\boldsymbol{Q}(Z|X)$ and the generating distribution $P_{\boldsymbol{C}}(U|Z)$, and Theorem 1 mainly shows how to factor it through $Z$.

In the first part, we will show that if $P_{\boldsymbol{C}}(U|Z)$ are Dirac measures, we have

$$\inf_{\Gamma\in\mathcal{P}(X\sim P_X,U\sim P_{\boldsymbol{C}})} \mathbb{E}_{(X,U)\sim\Gamma}\left\{\ell(f(X),U)\right\}$$
$$= \inf_{\Gamma\in\mathcal{P}_{X,U}} \mathbb{E}_{(X,U)\sim\Gamma}\left\{\ell(f(X),U)\right\}, \tag{4}$$

where $\mathcal{P}(X\sim P_X,U\sim P_{\boldsymbol{C}})$ denotes the set of all joint distributions of $(X,U)$ with marginals $P_X, P_{\boldsymbol{C}}$, and likewise for $\mathcal{P}(X\sim P_X,Z\sim P_Z)$. The set of all joint distributions of $(X,U,Z)$ such that $X\sim P_X$, $(U,Z)\sim P_{\boldsymbol{C},Z}$, and $(U\perp\!\!\!\perp X)|Z$ are denoted by $\mathcal{P}_{X,U,Z}$. $\mathcal{P}_{X,U}$ and $\mathcal{P}_{X,Z}$ denote the sets of marginals on $(X,U)$ and $(X,Z)$ induced by $\mathcal{P}_{X,U,Z}$.

From the definition, it is clear that $\mathcal{P}_{X,U}\subseteq\mathcal{P}(P_X,P_{\boldsymbol{C}})$. Therefore, we have

$$\inf_{\Gamma\in\mathcal{P}(X\sim P_X,U\sim P_{\boldsymbol{C}})} \mathbb{E}_{(X,U)\sim\Gamma}\left\{\ell(f(X),U)\right\}$$
$$\leq \inf_{\Gamma\in\mathcal{P}_{X,U}} \mathbb{E}_{(X,U)\sim\Gamma}\left\{\ell(f(X),U)\right\}, \tag{5}$$

The identity is satisfied if $P_{\boldsymbol{C}}(U|Z)$ are Dirac measures, such as $U = \boldsymbol{C}(Z)$. This is proved by the following Lemma in (Tolstikhin et al., 2017). -5pt

**Lemma 1** $\mathcal{P}_{X,U}\subseteq\mathcal{P}(P_X,P_{\boldsymbol{C}})$ *with identity if* $P_{\boldsymbol{C}}(U|Z=z)$ *are Dirac for all* $z\in\mathcal{Z}$. *(see details in (Tolstikhin et al., 2017).)*

In the following part, we show that

$$\inf_{\Gamma\in\mathcal{P}_{X,U}} \mathbb{E}_{(X,U)\sim\Gamma}\left\{\ell(f(X),U)\right\}$$
$$= \inf_{\boldsymbol{Q}:\boldsymbol{Q}_Z=P_Z} \mathbb{E}_{P_X}\mathbb{E}_{\boldsymbol{Q}(Z|X)}\left\{\ell(f(X),\boldsymbol{C}(Z))\right\}. \tag{6}$$

Based on the definition, $\mathcal{P}(P_X, P_C)$, $\mathcal{P}_{X,U,Z}$ and $\mathcal{P}_{X,U}$ depend on the choice of conditional distributions $P_C(U|Z)$, but $\mathcal{P}_{X,Z}$ does not. It is also easy to check that $\mathcal{P}_{X,Z} = \mathcal{P}(X \sim P_X, Z \sim P_Z)$. The tower rule of expectation, and the conditional independence property of $\mathcal{P}_{X,U,Z}$ implies

$$\inf_{\Gamma \in \mathcal{P}_{X,U}} \mathbb{E}_{(X,U) \sim \Gamma} \{\ell(f(X), U)\}$$

$$= \inf_{\Gamma \in \mathcal{P}_{X,U,Z}} \mathbb{E}_{(X,U,Z) \sim \Gamma} \{\ell(f(X), U)\}$$

$$= \inf_{\Gamma \in \mathcal{P}_{X,U,Z}} \mathbb{E}_{P_Z} \mathbb{E}_{X \sim P(X|Z)} \mathbb{E}_{U \sim P(U|Z)} \{\ell(f(X), U)\}$$

$$= \inf_{\Gamma \in \mathcal{P}_{X,U,Z}} \mathbb{E}_{P_Z} \mathbb{E}_{X \sim P(X|Z)} \{\ell(f(X), \boldsymbol{C}(Z))\}$$

$$= \inf_{\Gamma \in \mathcal{P}_{X,Z}} \mathbb{E}_{(X,Z) \sim \Gamma} \{\ell(f(X), \boldsymbol{C}(Z))\}$$

$$= \inf_{\boldsymbol{Q}:\boldsymbol{Q}_Z = P_Z} \mathbb{E}_{P_X} \mathbb{E}_{\boldsymbol{Q}(Z|X)} \{\ell(f(X), \boldsymbol{C}(Z))\} \tag{7}$$

Finally, since $Y = f(X)$, it is easy to get

$$\inf_{\Gamma \in \mathcal{P}(Y \sim P_Y, U \sim P_C)} \mathbb{E}_{(Y,U) \sim \Gamma} \{\ell(Y, U)\}$$

$$= \inf_{\Gamma \in \mathcal{P}(X \sim P_X, U \sim P_C)} \mathbb{E}_{(X,U) \sim \Gamma} \{\ell(f(X), U)\} \tag{8}$$

Now (4), (6) and (8) are proved and the three together prove Theorem 1.

Our proposed framework readily applies to non-deterministic case. If the classifier part is non-deterministic, Lemma 1 provides only the inclusion of sets $\mathcal{P}_{X,U} \subseteq \mathcal{P}(P_X, P_U)$, and we can get an upper bound on the Wasserstein distance between the ground-truth and predicted label distributions:

$$\inf_{\Gamma \in \mathcal{P}(X \sim P_X, U \sim P_C)} \mathbb{E}_{(X,U) \sim \Gamma} \{\ell(f(X), U)\} \leq \inf_{\Gamma \in \mathcal{P}_{X,U}} \mathbb{E}_{(X,U) \sim \Gamma} \{\ell(f(X), U)\}$$

$$\leq \sum_{i=1}^{d} \sigma_i^2 + \inf_{\Gamma \in \mathcal{P}_{X \sim P_X, Z \sim P_Z}} \mathbb{E}_{(X,Z) \sim \Gamma} \left\{\|f(X) - \boldsymbol{C}(Z)\|^2\right\}, \tag{9}$$

where we assume the conditional distributions $P_C(U|Z = z)$ have mean values $\boldsymbol{C}(z) \in \mathbb{R}^d$ and marginal variances $\sigma_1^2, ..., \sigma_d^2 \geq 0$ for all $z \in \mathcal{Z}$, where $\boldsymbol{C} : \mathcal{Z} \rightarrow \mathcal{X}$, and $\ell(y, u) = \|y - u\|^2$. The above upper bound is derived by:

$$\inf_{\Gamma \in \mathcal{P}_{X,U}} \mathbb{E}_{(X,U) \sim \Gamma} \left\{\|f(X) - U\|^2\right\} = \inf_{\Gamma \in \mathcal{P}_{X,U,Z}} \mathbb{E}_{P_Z} \mathbb{E}_{X \sim P(X|Z)} \mathbb{E}_{U \sim P(U|Z)} \{\|f(X) - U\|^2\} \tag{10}$$

and

$$\mathbb{E}_{U \sim P(U|Z)} \{\|f(X) - U\|^2\} = \mathbb{E}_{U \sim P(U|Z)} \{\|f(X) - \boldsymbol{C}(Z) + \boldsymbol{C}(Z) - U\|^2\}$$

$$= \|f(X) - \boldsymbol{C}(Z)\|^2 + \mathbb{E}_{U \sim P(U|Z)} \{< f(X) - \boldsymbol{C}(Z), \boldsymbol{C}(Z) - U >\} + \mathbb{E}_{U \sim P(U|Z)} \{\|\boldsymbol{C}(Z)| - U\|^2\}$$

$$= \|f(X) - \boldsymbol{C}(Z)\|^2 + \sum_{i=1}^{d} \sigma_i^2. \tag{11}$$

In equation 11, the second term of the second last row becomes 0 since the optimization will drive $f(X) - \boldsymbol{C}(Z)$ to zero.

## HYPER-PARAMETER SELECTION

### DIMENSION OF EMBEDDING SPACE

One important hyper-parameter for the ER-Classifier is the dimension of the embedding space. If the dimension is too small, important features are "collapsed" onto the same dimension, and if the dimension is too large, the projection will not extract useful information, which results in too much noise and instability. The maximum likelihood estimation of intrinsic dimension proposed in

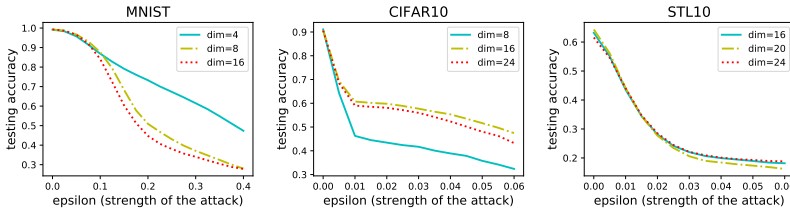

Figure 5: Testing accuracy of models with different embedding dimensions under $l_\infty$-PGD attack.

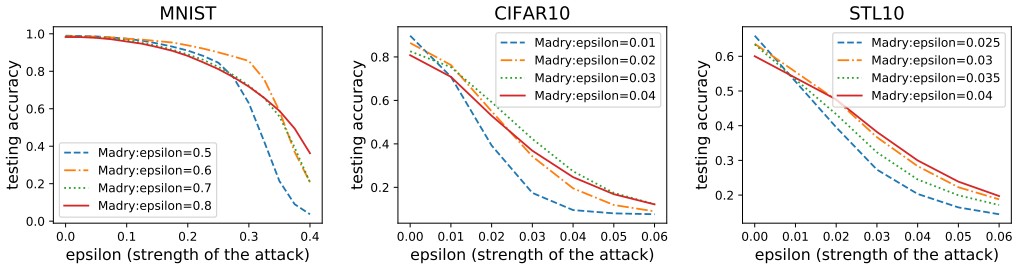

Figure 6: Testing accuracy of models with different $\epsilon$ on MNIST, CIFAR10 and STL10.

(Levina & Bickel, 2005)[3] is used to calculate the intrinsic dimension of each image dataset, serving as a guide for selecting the embedding dimension. The sample size used in calculating the intrinsic dimension is $1,000$, and changing the sample size does not influence the results much. Based on the intrinsic dimension calculated by (Levina & Bickel, 2005), we test several different values around the suggested intrinsic dimension and evaluate the models against $l_\infty$-PGD attack. All models are trained without min-max robust optimization, and the experimental results are shown in Figure 5.

The final embedding dimension is chosen based on robustness, number of parameters, and testing accuracy when there is no attack. The final embedding dimensions and suggested intrinsic dimensions are shown in Table 3.

| Data | Data dim. | Intrinsic dim. | Embedding dim. |
|---|---|---|---|
| MNIST | $1 \times 28 \times 28$ | 13 | 4 |
| CIFAR10 | $3 \times 32 \times 32$ | 17 | 16 |
| STL10 | $3 \times 96 \times 96$ | 20 | 16 |
| Tiny Imagenet | $3 \times 64 \times 64$ | 19 | 20 |

Table 3: Pixel space dimension, intrinsic dimension calculated by (Levina & Bickel, 2005), and final embedding dimension used.

Based on Figure 5, the embedding dimension close to the calculated intrinsic dimension usually offers better results except on MNIST. One explanation may be that MNIST is a simple handwritten digit dataset, so performing classification on MNIST may not require that many dimensions.

EPSILON SELECTION

Epsilon ($\epsilon$) is an important hyper-parameter for adversarial training. When doing Madry's adversarial training, we test the model robustness with different $\epsilon$ and choose the best one. The experiment results are shown in Figure 6.

Based on Figure 6, we use $\epsilon = 0.3, 0.03, 0.03$ in Madry's adversarial training on MNIST, CIFAR10 and STL10 respectively. For Tiny Imagenet, we use $\epsilon = 0.01$. To make a fair comparison, we use the same $\epsilon$ when training ER-Classifier.

---

[3]Code publicly available at `https://github.com/OFAI/hub-toolbox-python3`

## EMBEDDING VISUALIZATION

In this section, we compare the embedding learned by Encoder+Classifier structure (E-CLA) and the embedding learned by ER-Classifier on several datasets without min-max robust optimization. We first generate embedding of testing data using the encoder ($\tilde{z} = \boldsymbol{Q}_\phi(x)$), then project the embedding points ($\tilde{z}$) to 2-D space by tSNE(Maaten & Hinton, 2008). Then we generate adversarial images ($x_{adv}$) against E-CLA and ER-Classifier using $l_\infty$-PGD attack. The adversarial embedding is generated by feeding the adversarial images into the encoder ($\tilde{z}_{adv} = \boldsymbol{Q}_\phi(x_{adv})$). Finally, we project the adversarial embedding points ($\tilde{z}_{adv}$) to 2-D space. The results are shown in Figure 7. The plots in the first and second rows are embedding visualization plots for E-CLA, and the plots in the third and last rows are the embedding visualization plots for ER-Classifier. In adversarial embedding visualization plots, the misclassified point is marked as "down triangle", which means the PGD attack successfully changed the prediction, and the correctly classified point is marked as "point", which means the attack fails.

Based on Figure 7, we can see that E-CLA can learn a good embedding on legitimate images of MNIST. Embedding points for different classes are separated on the 2D space, but under adversarial attack, some embedding points of different classes are mixed together. However, ER-Classifier can generate good separated embeddings on both legitimate and adversarial images. On CIFAR10, the E-CLA can not generate good separated embeddings on either legitimate images or adversarial images, while ER-Classifier can generate good separated embeddings for both.

## PSEUDO-CODE

Code for reproduction will be made available online at github later. The pseudocode for training ER-Classifier is shown in Listing 1.

## MODEL STRUCTURE

MNIST, STL10 and TinyImagenet classifier structures used for baseline methods are shown in Figure 8. We use VGG19 for the baseline methods on CIFAR10. Details of ER-Classifier structures on the four benchmark datasets are shown in Figure 9-10.

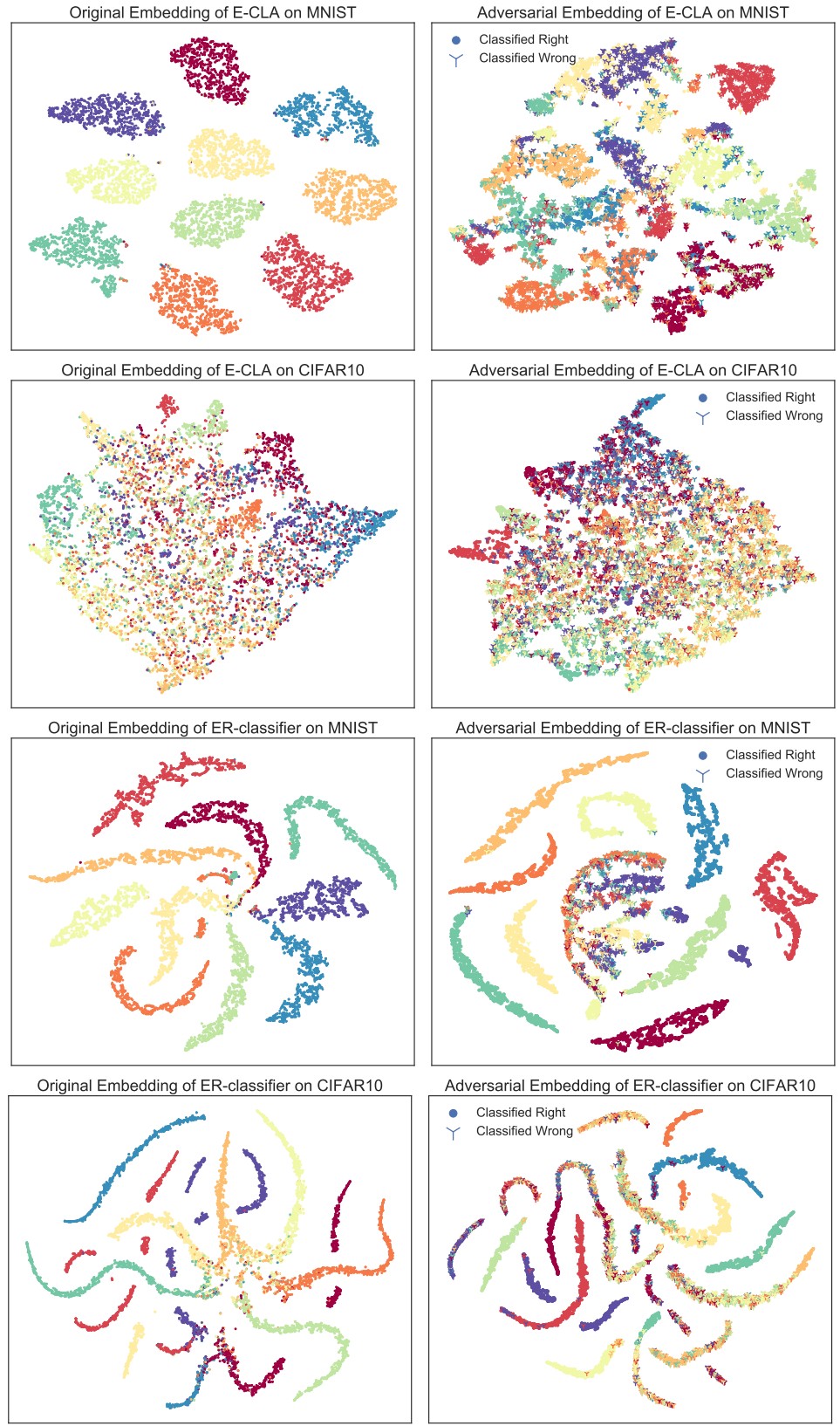

Figure 7: 2D embeddings for E-CLA and ER-Classifier on MNIST and CIFAR10. Larger visualization figures are also shown in the appendix.

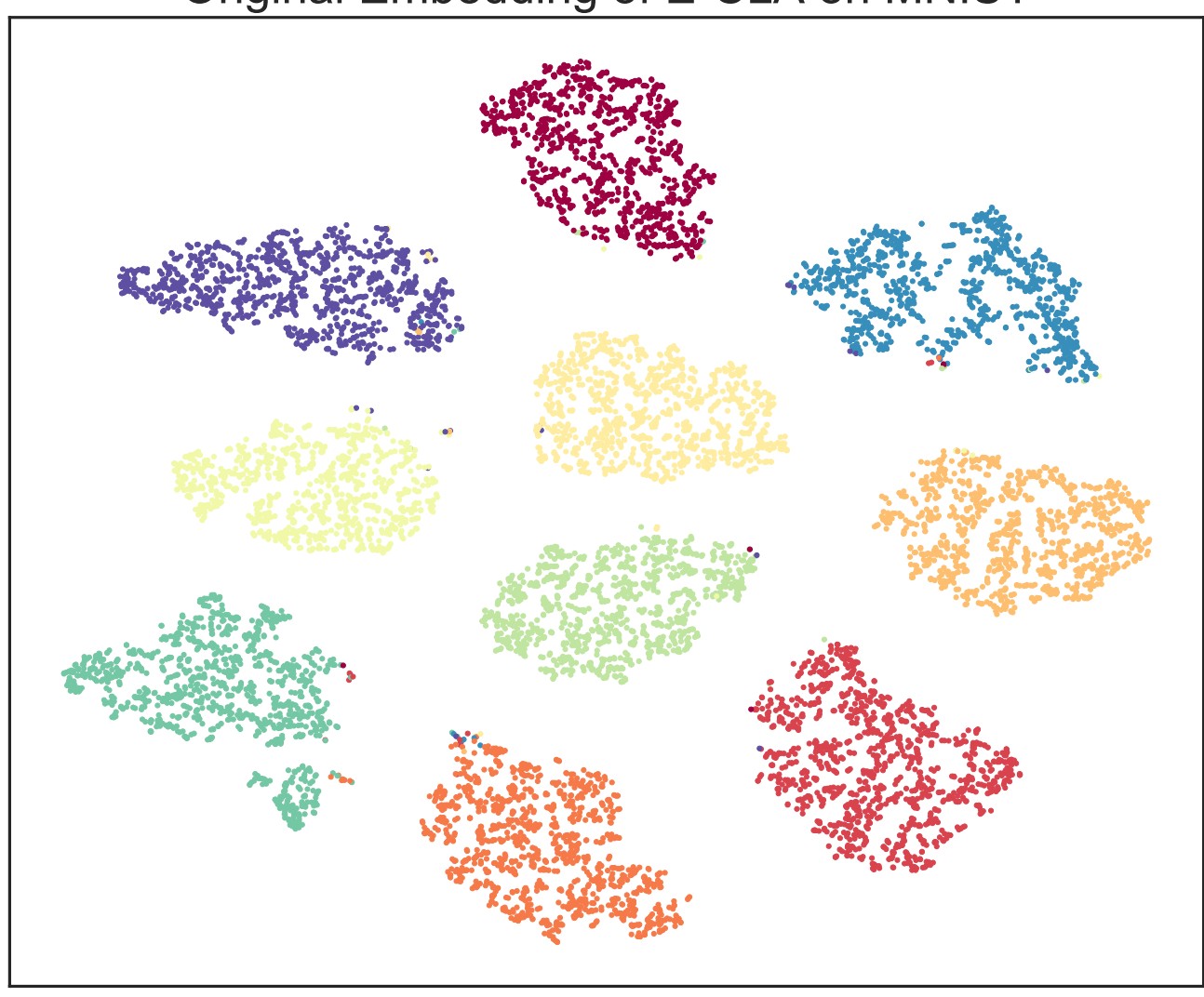

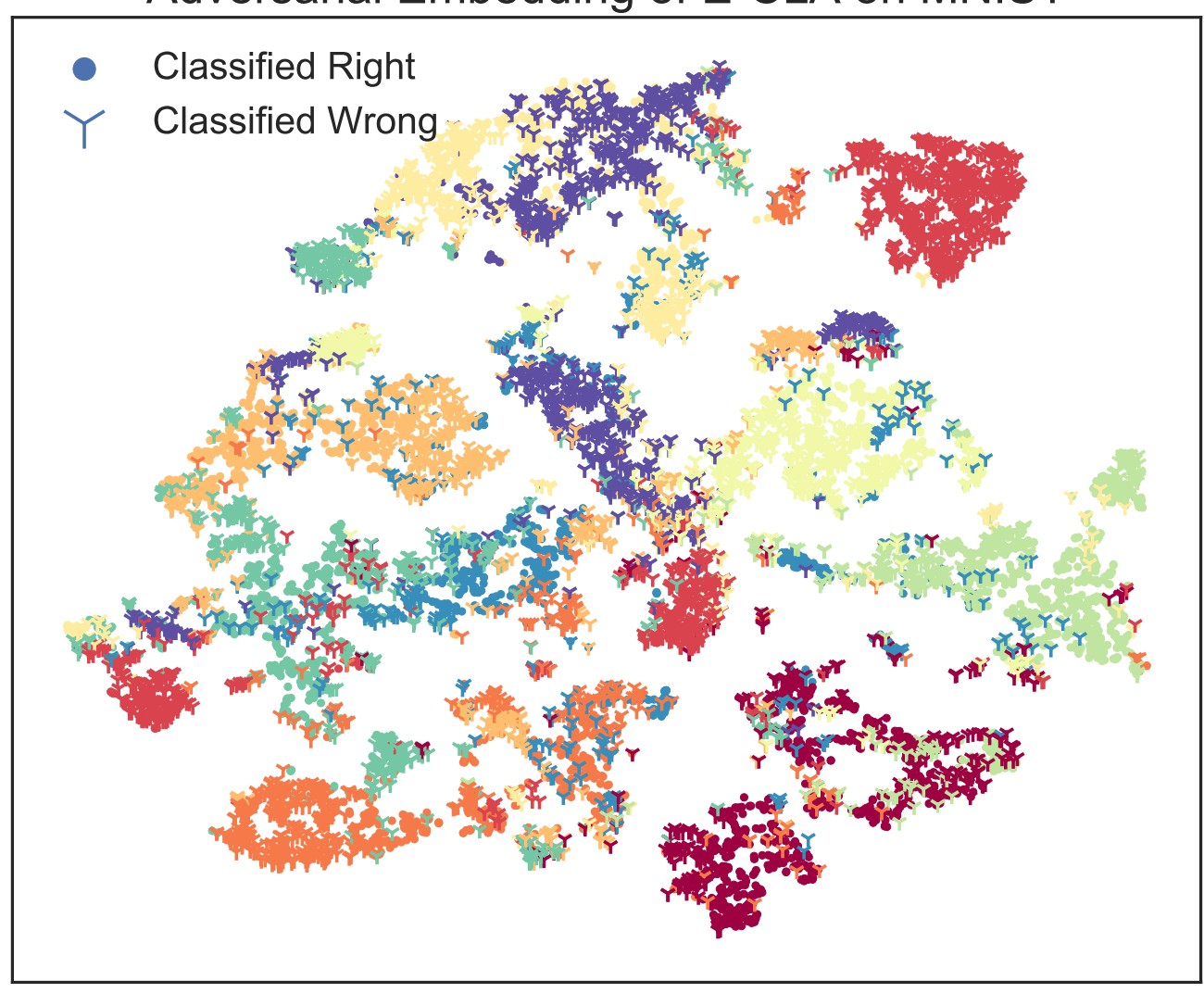

Adversarial Embedding of E-CLA on MNIST

# Original Embedding of E-CLA on CIFAR10

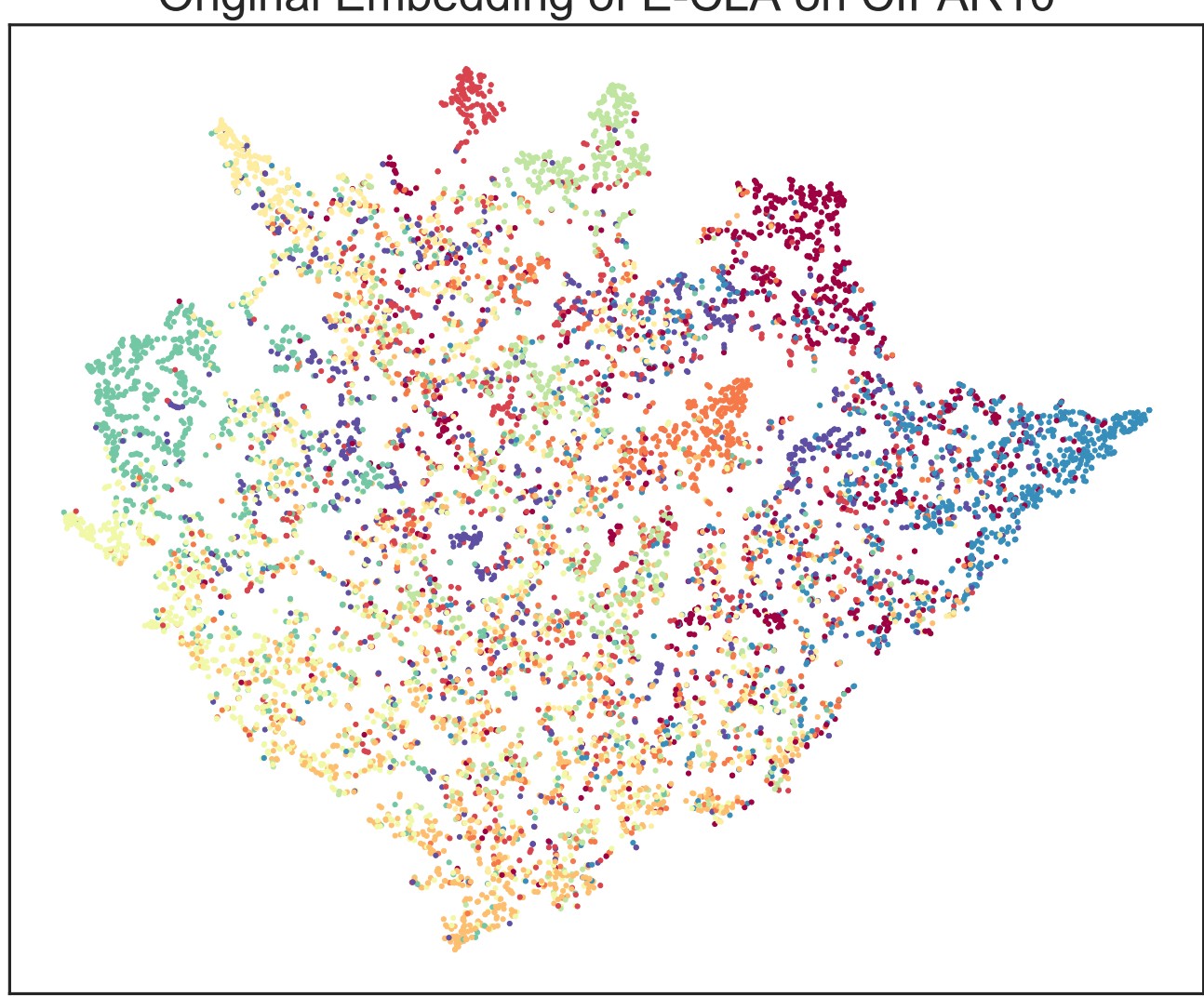

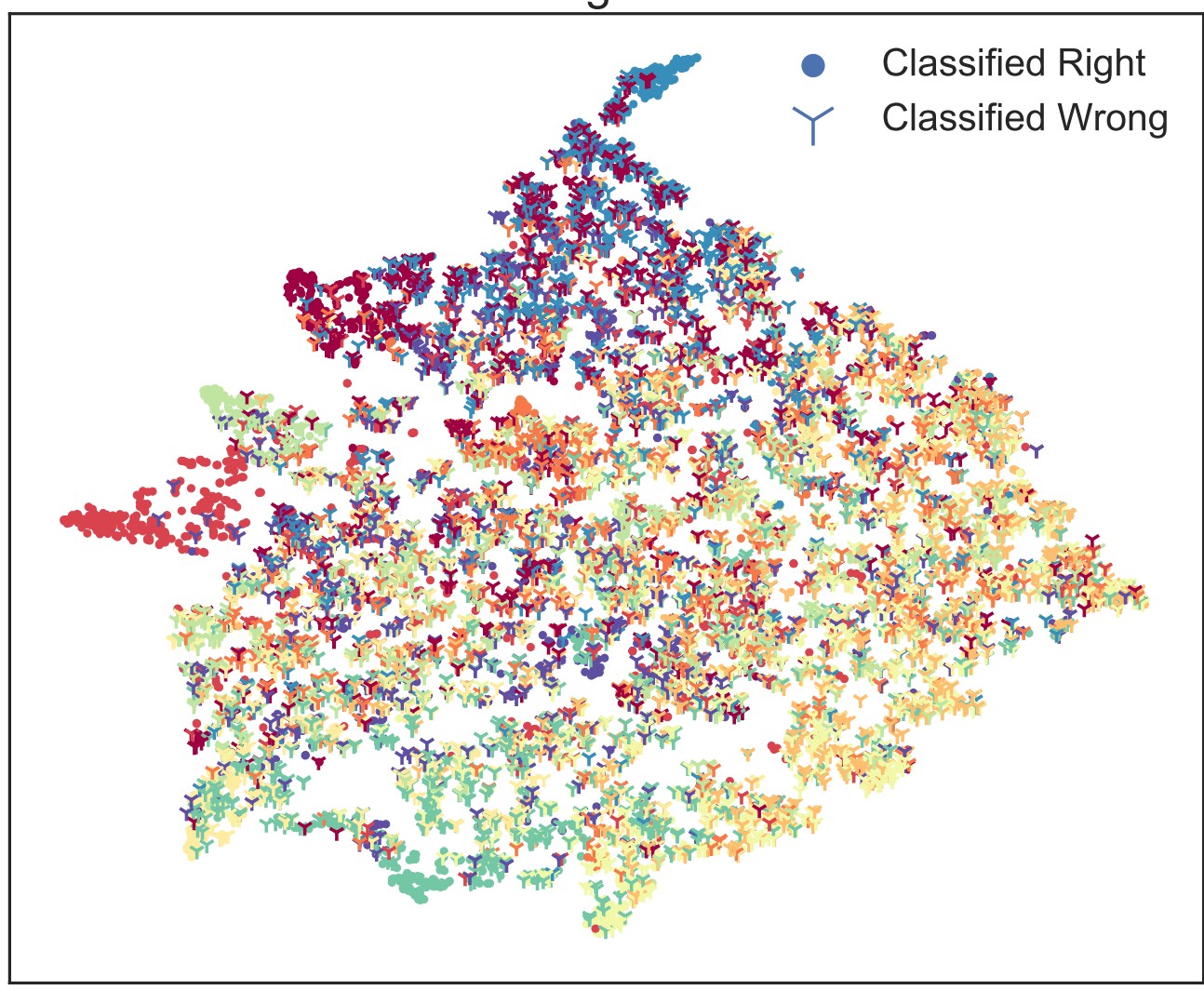

Original Embedding of ER-classifier on MNIST

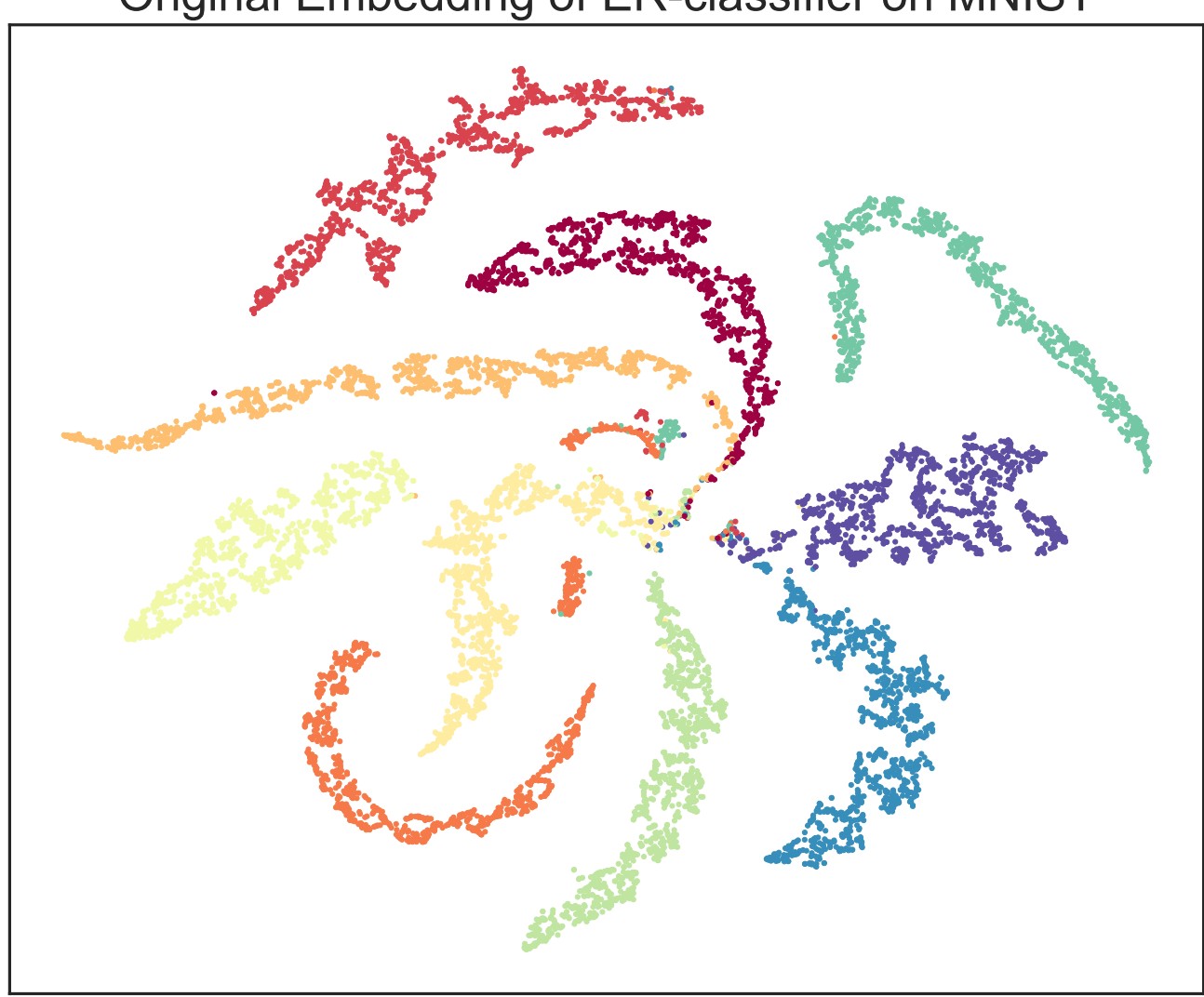

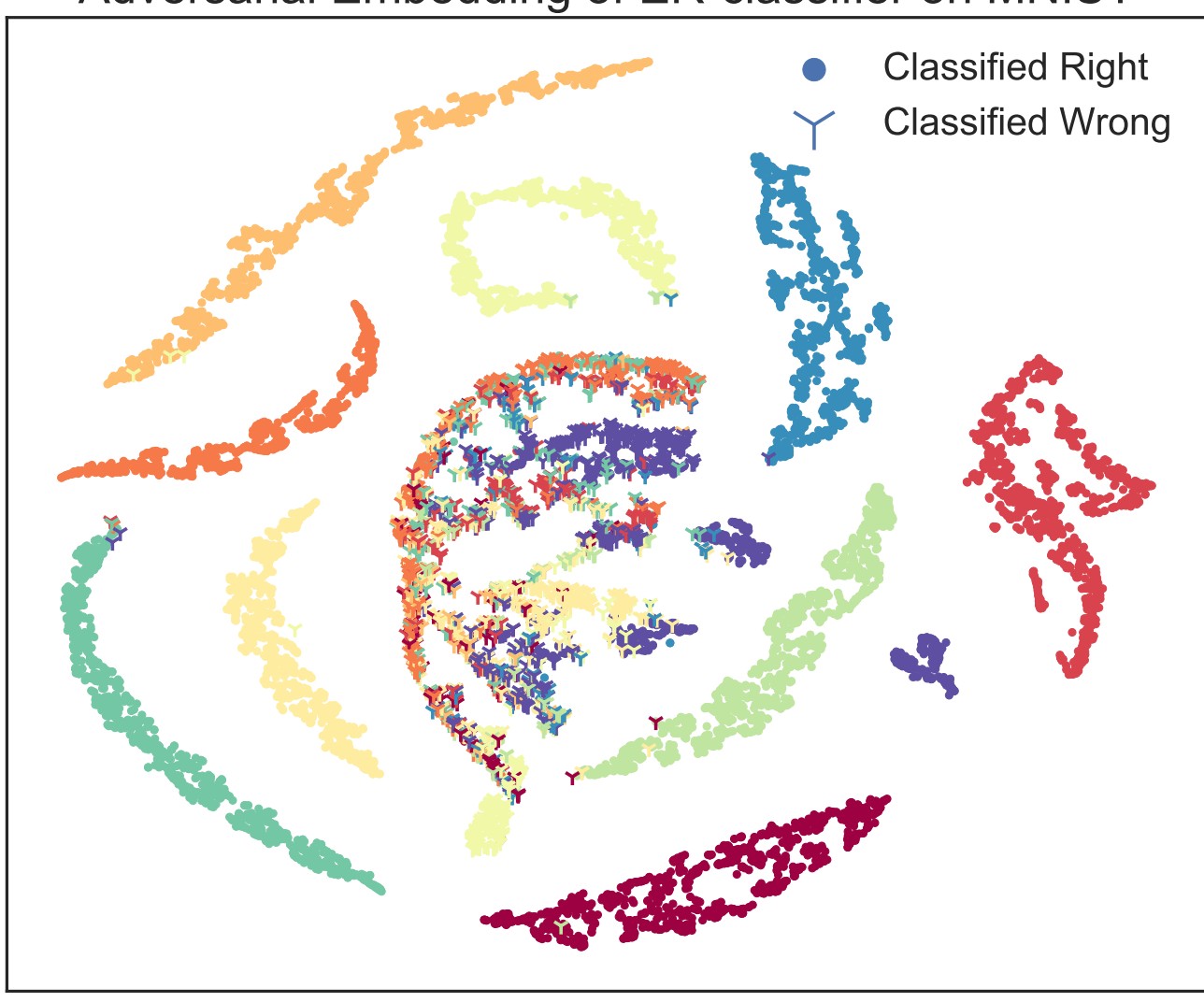

Original Embedding of ER-classifier on CIFAR10

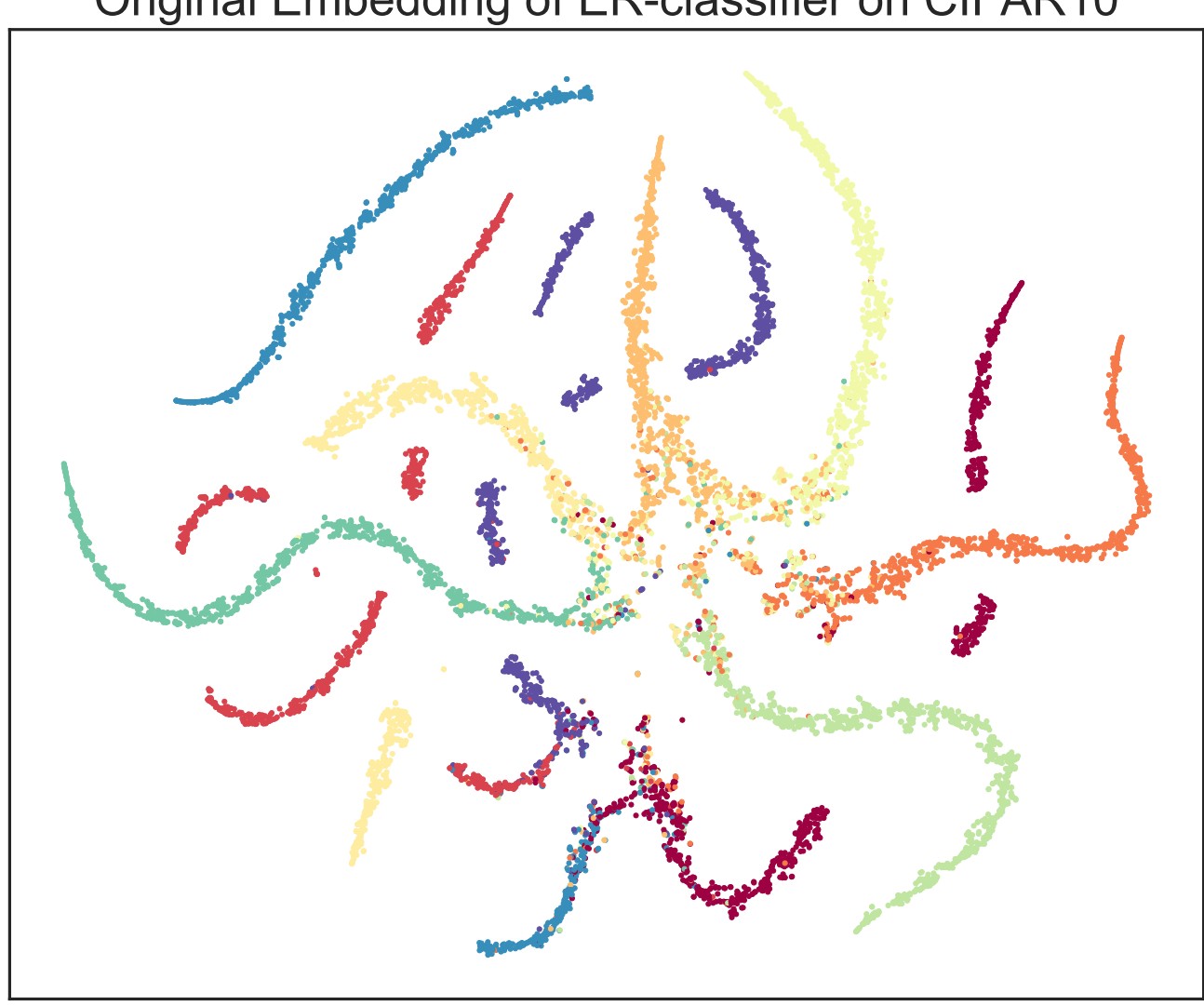

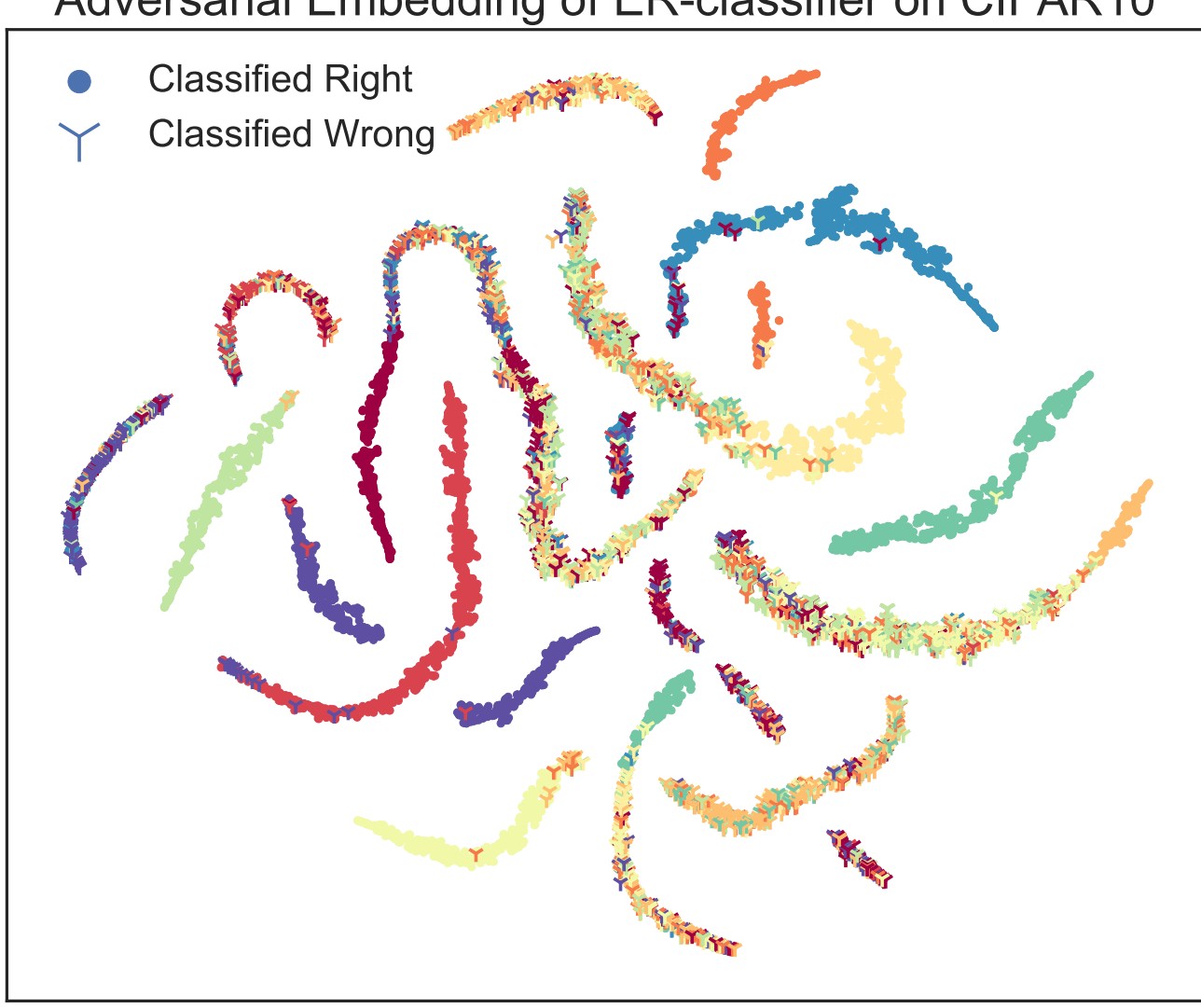

```python
def train_er_cla(train_loader, test_loader, encoder, discriminator,
                 classifier, other_hyper_parameters):
    criterion = nn.CrossEntropyLoss()
    encoder.train()
    discriminator.train()
    classifier.train()
    # Optimizers
    enc_optim = optim.Adam(encoder.parameters(), lr = lr)
    dis_optim = optim.Adam(discriminator.parameters(), lr = 0.5 * lr)
    cla_optim = optim.Adam(classifier.parameters(), lr = 0.05 * lr)
    enc_scheduler = StepLR(enc_optim, step_size=30, gamma=0.5)
    dis_scheduler = StepLR(dis_optim, step_size=30, gamma=0.5)
    cla_scheduler = StepLR(cla_optim, step_size=30, gamma=0.5)
    one = torch.Tensor([1])
    mone = one * -1
    for epoch in range(num_epoch):
        step = 0
        for images, labels in tqdm(train_loader):
            encoder.zero_grad()
            discriminator.zero_grad()
            classifier.zero_grad()
            # ======== Min-Max Robust Optimization ======== #
            images = adv_get(images, classifier, encoder)
            # ======== Train Discriminator ======== #
            frozen_params(encoder)
            frozen_params(classifier)
            free_params(discriminator)
            z_fake = sample_z(prior, n_z, batch_size, sigma)
            d_fake = discriminator(to_var(z_fake))
            z_real = encoder(images)
            d_real = discriminator(to_var(z_real))
            disc_fake = LAMBDA * d_fake.mean()
            disc_real = LAMBDA * d_real.mean()
            disc_fake.backward(one)
            disc_real.backward(mone)
            diss_loss = disc_fake - disc_real
            dis_optim.step()
            clip_params(discriminator)
            # ======== Train Classifier and Encoder======== #
            free_params(encoder)
            free_params(classifier)
            frozen_params(discriminator)
            pred_labels = classifier(encoder(to_var(images)))
            class_loss = LAMBDA0 * criterion(pred_labels, labels)
            class_loss.backward()
            cla_optim.step()
            enc_optim.step()
            # ======== Train Encoder ======== #
            free_params(encoder)
            frozen_params(classifier)
            frozen_params(discriminator)
            z_real = encoder(images)
            d_real = discriminator(encoder(Variable(images.data)))
            d_loss = LAMBDA1 * (d_real.mean())
            d_loss.backward(one)
            enc_optim.step()
            step += 1
    savefile(file_name, encoder, discriminator, classifier, dataset=
    dataset)
    return classifier, encoder
```

Listing 1: Pseudocode for training ER-Classifier

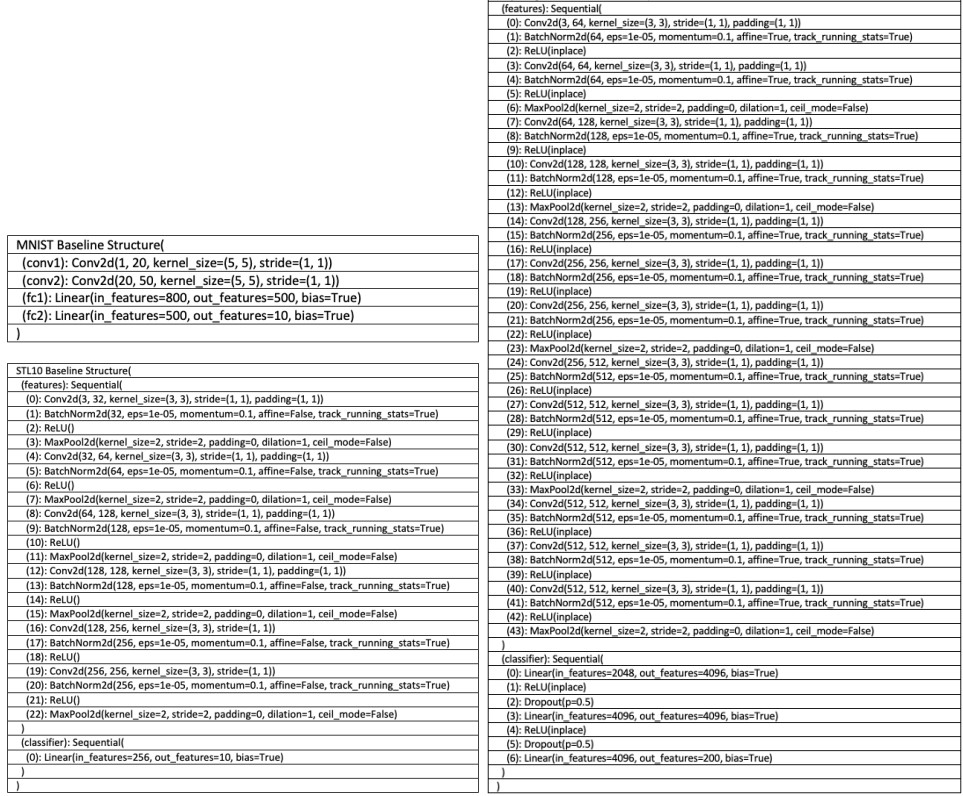

Figure 8: Baseline Structure for MNIST

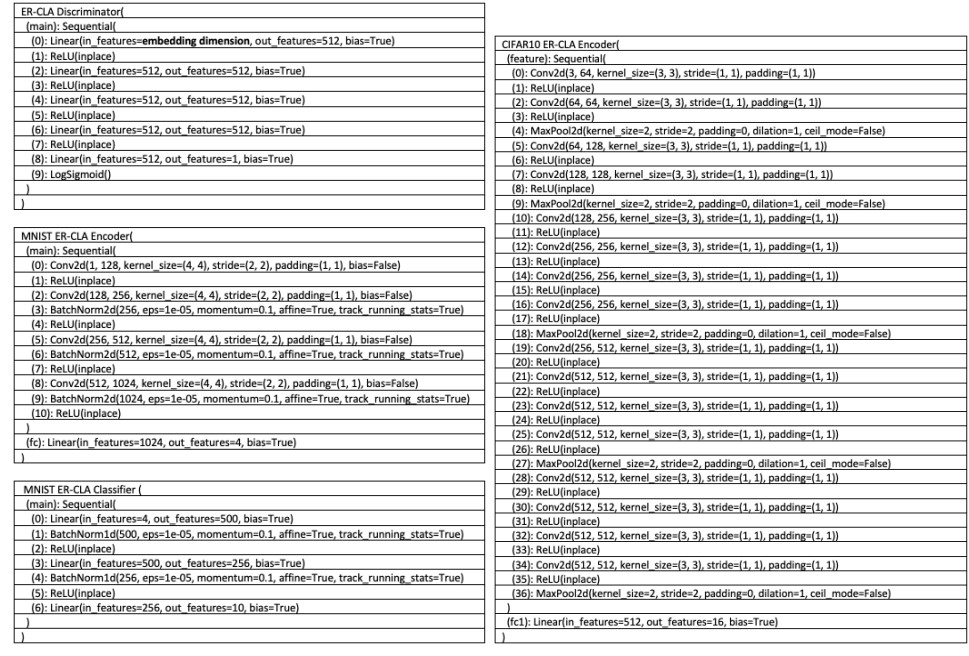

Figure 9: Details of Embedding Regularized Classifier Structures

```
Tiny ImageNet ER-CLA Encoder(
  (features): Sequential(
    (0): Conv2d(3, 64, kernel_size=(3, 3), stride=(1, 1), padding=(1, 1))
    (1): BatchNorm2d(64, eps=1e-05, momentum=0.1, affine=True, track_running_stats=True)
    (2): ReLU(inplace)
    (3): Conv2d(64, 64, kernel_size=(3, 3), stride=(1, 1), padding=(1, 1))
    (4): BatchNorm2d(64, eps=1e-05, momentum=0.1, affine=True, track_running_stats=True)
    (5): ReLU(inplace)
    (6): MaxPool2d(kernel_size=2, stride=2, padding=0, dilation=1, ceil_mode=False)
    (7): Conv2d(64, 128, kernel_size=(3, 3), stride=(1, 1), padding=(1, 1))
    (8): BatchNorm2d(128, eps=1e-05, momentum=0.1, affine=True, track_running_stats=True)
    (9): ReLU(inplace)
    (10): Conv2d(128, 128, kernel_size=(3, 3), stride=(1, 1), padding=(1, 1))
    (11): BatchNorm2d(128, eps=1e-05, momentum=0.1, affine=True, track_running_stats=True)
    (12): ReLU(inplace)
    (13): MaxPool2d(kernel_size=2, stride=2, padding=0, dilation=1, ceil_mode=False)
    (14): Conv2d(128, 256, kernel_size=(3, 3), stride=(1, 1), padding=(1, 1))
    (15): BatchNorm2d(256, eps=1e-05, momentum=0.1, affine=True, track_running_stats=True)
    (16): ReLU(inplace)
    (17): Conv2d(256, 256, kernel_size=(3, 3), stride=(1, 1), padding=(1, 1))
    (18): BatchNorm2d(256, eps=1e-05, momentum=0.1, affine=True, track_running_stats=True)
    (19): ReLU(inplace)
    (20): Conv2d(256, 256, kernel_size=(3, 3), stride=(1, 1), padding=(1, 1))
    (21): BatchNorm2d(256, eps=1e-05, momentum=0.1, affine=True, track_running_stats=True)
    (22): ReLU(inplace)
    (23): MaxPool2d(kernel_size=2, stride=2, padding=0, dilation=1, ceil_mode=False)
    (24): Conv2d(256, 512, kernel_size=(3, 3), stride=(1, 1), padding=(1, 1))
    (25): BatchNorm2d(512, eps=1e-05, momentum=0.1, affine=True, track_running_stats=True)
    (26): ReLU(inplace)
    (27): Conv2d(512, 512, kernel_size=(3, 3), stride=(1, 1), padding=(1, 1))
    (28): BatchNorm2d(512, eps=1e-05, momentum=0.1, affine=True, track_running_stats=True)
    (29): ReLU(inplace)
    (30): Conv2d(512, 512, kernel_size=(3, 3), stride=(1, 1), padding=(1, 1))
    (31): BatchNorm2d(512, eps=1e-05, momentum=0.1, affine=True, track_running_stats=True)
    (32): ReLU(inplace)
    (33): MaxPool2d(kernel_size=2, stride=2, padding=0, dilation=1, ceil_mode=False)
    (34): Conv2d(512, 512, kernel_size=(3, 3), stride=(1, 1), padding=(1, 1))
    (35): BatchNorm2d(512, eps=1e-05, momentum=0.1, affine=True, track_running_stats=True)
    (36): ReLU(inplace)
    (37): Conv2d(512, 512, kernel_size=(3, 3), stride=(1, 1), padding=(1, 1))
    (38): BatchNorm2d(512, eps=1e-05, momentum=0.1, affine=True, track_running_stats=True)
    (39): ReLU(inplace)
    (40): Conv2d(512, 512, kernel_size=(3, 3), stride=(1, 1), padding=(1, 1))
    (41): BatchNorm2d(512, eps=1e-05, momentum=0.1, affine=True, track_running_stats=True)
    (42): ReLU(inplace)
    (43): MaxPool2d(kernel_size=2, stride=2, padding=0, dilation=1, ceil_mode=False)
  )
  (avgpool): AdaptiveAvgPool2d(output_size=1)
  (fc1): Linear(in_features=512, out_features=20, bias=True)
)
```

```
CIFAR10 ER-CLA Classifier(
  (main): Sequential(
    (0): Linear(in_features=16, out_features=512, bias=True)
    (1): Dropout(p=0.5)
    (2): Linear(in_features=512, out_features=512, bias=True)
    (3): ReLU(inplace)
    (4): Dropout(p=0.5)
    (5): Linear(in_features=512, out_features=512, bias=True)
    (6): ReLU(inplace)
    (7): Linear(in_features=512, out_features=10, bias=True)
  )
)
```

```
Tiny ImageNet ER-CLA Classifier(
  (main): Sequential(
    (0): Linear(in_features=20, out_features=4096, bias=True)
    (1): ReLU(inplace)
    (2): Dropout(p=0.5)
    (3): Linear(in_features=4096, out_features=4096, bias=True)
    (4): ReLU(inplace)
    (5): Dropout(p=0.5)
    (6): Linear(in_features=4096, out_features=200, bias=True)
  )
)
```

```
STL10 ER-CLA Encoder(
  (main1): Sequential(
    (0): Conv2d(3, 128, kernel_size=(4, 4), stride=(2, 2), padding=(1, 1), bias=False)
    (1): ReLU(inplace)
    (2): Conv2d(128, 256, kernel_size=(4, 4), stride=(2, 2), padding=(1, 1), bias=False)
    (3): BatchNorm2d(256, eps=1e-05, momentum=0.1, affine=True, track_running_stats=True)
    (4): ReLU(inplace)
    (5): Conv2d(256, 512, kernel_size=(4, 4), stride=(2, 2), padding=(1, 1), bias=False)
    (6): BatchNorm2d(512, eps=1e-05, momentum=0.1, affine=True, track_running_stats=True)
    (7): ReLU(inplace)
    (8): Conv2d(512, 1024, kernel_size=(4, 4), stride=(2, 2), padding=(1, 1), bias=False)
    (9): BatchNorm2d(1024, eps=1e-05, momentum=0.1, affine=True, track_running_stats=True)
    (10): ReLU(inplace)
    (11): Conv2d(1024, 2048, kernel_size=(4, 4), stride=(2, 2), padding=(1, 1), bias=False)
    (12): BatchNorm2d(2048, eps=1e-05, momentum=0.1, affine=True, track_running_stats=True)
    (13): ReLU(inplace)
    (14): Conv2d(2048, 2048, kernel_size=(3, 3), stride=(2, 2), bias=False)
    (15): BatchNorm2d(2048, eps=1e-05, momentum=0.1, affine=True, track_running_stats=True)
    (16): ReLU(inplace)
  )
  (fc1): Linear(in_features=2048, out_features=16, bias=True)
)
```

```
STL10 ER-CLA Classifier(
  (main): Sequential(
    (0): Linear(in_features=16, out_features=500, bias=True)
    (1): BatchNorm1d(500, eps=1e-05, momentum=0.1, affine=True, track_running_stats=True)
    (2): ReLU(inplace)
    (3): Linear(in_features=500, out_features=256, bias=True)
    (4): BatchNorm1d(256, eps=1e-05, momentum=0.1, affine=True, track_running_stats=True)
    (5): ReLU(inplace)
    (6): Linear(in_features=256, out_features=10, bias=True)
  )
)
```

Figure 10: Details of Embedding Regularized Classifier Structures

