# OpenReview forum: "Defending Against Adversarial Examples by Regularized Deep Embedding"
_ICLR.cc/2020/Conference — Reject_

### Official Review · AnonReviewer1 · 2019-10-16
**Official Blind Review #1**

**Rating:** 1

**Review:**


Summary
========
This paper proposes to make neural networks robust to adversarial examples via dimensionality reduction.
The paper builds on and compares to prior approaches (e.g., MagNet and Defense-GAN) that are known to be broken, which casts serious doubts on the validity of the performed experimental evaluation.
The theoretical part of the paper is also very hard to follow.
For these reasons, I recommend rejection.


Comments
=========

The way I understand the proposed framework, it is trying to compress the data into a form that is as close as possible to a standard Gaussian distribution, while maintaining classification accuracy.
The intuition, formulated next to Figure 1, seems to be that compressing the data will force the network to only retain the most important features for classification, which will thus lead to higher robustness.
I have two main concerns with this approach:
- First, recent work by Ilyas et al ("Adversarial examples are not bugs, they are features") seems to suggest that models tend to learn features that are not robust yet generalize well. It is unclear why compressing the data would remove such features.
- Second, it is unclear why constraining the compressed data to be close to a Gaussian would have a positive effect on robustness. The authors claim that this is to remove some "pathological" mappings from the input space to the embedding space. Can you explain what you mean here? What are these pathological mappings and why should they be the source of non-robustness?

The theoretical analysis in Sections 2.3and 3. is very hard to follow. The authors first mention that "optimal transport theory [...] provides a much weaker topology than many other [distances]". What do you mean by this? What is the advantage of optimal transport theory that you are trying to exploit?
When Kantorovich’s distance is first introduced, the reader has no idea what any of the symbols represent. What are Y, U, \mathcal{U}, P_Y, P_C, etc?
The authors then mention a relation to the Wasserstein distance and to Wasserstein GANs. How does this relate to this paper?

I could not understand how Algorithm 1 related to the discussion in Section 3.
Algorithm 1 simply uses a standard cross-entropy loss to train the classifier C. Is this what you mean by minimizing the optimal transport cost between P_Y and P_C? This seems like a very convoluted way of justifying the use of the most standard loss function in ML.
Algorithm 1, step 6 mentions sampling from Q(Z|x_i). But isn't the encoder Q deterministic? If not, where does the randomness come from?

In the experimental section, the authors compare to Defense-GAN, which was shown to be broken in "The Robust Manifold Defense: Adversarial Training using Generative Models" by Jalal et al.
The authors mention that they evaluate robustness using an untargeted white-box PGD attack, but they do not specify which objective their attack is optimizing. As this papers proposed a different classification pipeline, adaptive the attacks to this pipeline is crucial.

The experimental results (Figure 2 & 3) raise a number of questions:
- Why does the ER-classifier variant without adversarial training perform so much better than the one with? The author's explanation about the difficulty of optimizing over the embedding space should be supported.
- Do the attacks reach 0% accuracy for large enough epsilon? The very high accuracy even for large epsilon suggests that the evaluated attacks are not evaluating the right objective to fool the classifier. E.g., on MNIST, eps=0.4 can nearly destroy the whole image so the accuracy should be expected to be lower.

**Experience Assessment:**

I have published in this field for several years.

**Review Assessment: Checking Correctness Of Derivations And Theory:**

I carefully checked the derivations and theory.

**Review Assessment: Checking Correctness Of Experiments:**

I carefully checked the experiments.

**Review Assessment: Thoroughness In Paper Reading:**

I read the paper thoroughly.

---

> ### Author Response · Authors · 2019-11-06
> **Response to Official Blind Review #1 (Part 2)**
>
> (3) Theoretical part and Wasserstein distance.
>
> 3.0 "The theoretical part of the paper is also very hard to follow. For these reasons, I recommend rejection."
>
> This is a very good suggestion. We agree that we should improve the presentation of the theory part. We will significantly improve it in revision. However, we disagree that this small presentation issue should get a strong reject, which will be further explained in 3.1 and 3.2.
>
> 3.1 "why Algorithm 1 related to the theory discussion in Section 3"
>
> There are two Wasserstein distances (W-distances) in our framework. One is the W-distance  between the latent distribution and the prior distribution, and the other one is the W-distance between the label distribution and the framework output distribution. In the algorithm, we are minimizing the first one. The theory shows that minimizing the first W-distance in combination with minimizing a standard cross-entropy loss is equivalent to minimizing the second W-distance, which guarantees that the training process is not distracted from the main goal of the framework, classification. We admit that putting this theory part before illustrating Algorithm is not easy for reviewers to follow. We improved the presentation and moved the complete theoretical analysis to the Appendix.
>
> Though deterministic encoder is used in the experiment, the structure can also be stochastic (like VAE that learns a stochastic encoder from which we can sample with the learned mean and variance).  In Appendix, we also provided a bound for the stochastic case.  Sampling from deterministic encoder just means readily getting the deterministic latent embeddings, which is similar to the deterministic encoder used in Wasserstein Autoencoder. We addressed this in the revised pdf.
>
> 3. 2 "optimal transport theory [...] provides a much weaker topology than many other [distances]"
>
> This term has been used and clearly explained in Wasserstein GAN [1]. Generative models are trying to learn a probability distribution. Various distances can be applied to measure the divergence between the model distribution and the real distribution. The difference between Wasserstein distance and some other  strong distances is the impact on the convergence of sequences of probability distributions. Informally, a distance induces a weaker topology when it makes it easier for a sequence of distribution to converge [1]. The weak distance makes it easier to define a continuous mapping from parametric space (theta-space) to the model density space (P_theta-space).  When training the generative model, we'd like to have a continuous loss function on theta to dist(P_theta, P_real-data). One practical benefit of 1-Wasserstein distance is the ability to continuously estimate the distance by training the discriminator to optimality. More explanation can be found in [1].
>
> (4) Experiments
>
> 4.1 "The paper builds on and compares to prior approaches (e.g., MagNet and Defense-GAN) that are known to be broken, which casts serious doubts on the validity of the performed experimental evaluation."
>
> First of all, we did not compare with MagNet.  Second, the reason we compared with Defense-GAN is that we share the similar idea of projecting the input to a learned manifold.
>
> More importantly, Defense-GAN is not the main baseline we'd like to compare with since it is shown to be partly broken in [2]. The main baseline is Madry's adversarial training, which has been shown to be better than Defense-GAN in [2].
>
> Removing the comparisons with Defense-GAN does not affect the contribution of our paper at all, because it's only 1% of the whole experimental results --- only a tiny portion in Table 2.
>
> The objective of the untargeted white-box attack is the untargeted loss of CW attack. We adapted the code from the github of [2].
>
> 4.2 "Other questions".
>
> ER-Classifier- only performs better than ER-Classifier when epsilon is large on CIFAR10. Reviewer 3 also mentioned this situation. Though we are also not clear about the reason, we provided our thoughts in the previous reply. We evaluated defense methods on MNIST with PGD. In testing phase, only classification loss is considered. The gradient used is the gradient of classification loss w.r.t the input. We have visualized the generated adversarial examples with different epsilon. People can still recognize the corresponding digit with large epsilon=0.4.
>
> Our empirical results along with the embedding visualization comparisons are very impressive, which we believe is inspiring and deserves more discussions from the community.
>
> [1] Wasserstein GAN, Martin Arjovsky, Soumith Chintala, Léon Bottou
> [2] Obfuscated Gradients Give a False Sense of Security: Circumventing Defenses to Adversarial Examples, Anish Athalye, Nicholas Carlini, David Wagner

---

> > ### Author Response · Authors · 2019-11-07
> > **Response to Official Blind Review #1 (Part 4)**
> >
> > Additional discussion for "The theoretical part of the paper is also very hard to follow.":
> >
> > The notations criticized by the reviewer are about standard definitions of optimal transport cost between two distributions. We will make them clearer in revision. As we explained in Part 2:  "There are two Wasserstein distances (W-distances) in our framework. One is the W-distance  between the latent distribution and the prior distribution, and the other one is the W-distance between the label distribution and the framework output distribution. In the algorithm, we are minimizing the first one. The theory shows that minimizing the first W-distance in combination with minimizing a standard cross-entropy loss is equivalent to minimizing the second W-distance, which guarantees that the training process is not distracted from the main goal of the framework, classification."
> >
> > Intuitively, this theory simply says that, if we regularize the marginal distribution of the low-dimensional embedding z that is the output of the encoder, q(z|x) with x integrated out, through minimizing an optimal transport cost (the first W-distance), we will guarantee that minimizing standard cross-entropy classification error will result in a classifier with the following property: the global output distribution of the classifier will match the global ground-truth label distribution in the dataset NO MATTER WHETHER THE ENCODER Q(Z|X) IS DETERMINISTIC OR STOCHASTIC (the second W-distance is automatically minimized).
> >
> > At first glance, this theory seems trivial and obvious. If we simply minimize the cross-entropy loss without any regularization, is it obvious that the global marginal label distribution in the dataset is also exactly preserved? Why preserving global label distribution helps? We can have a wrong classifier that always outputs the wrong label for each data point but preserves the global label frequency in the dataset.
> >
> > In fact, the theory is NEITHER TRIVIAL NOR OBVIOUS. It's hard to analyze its importance if we just look at a deterministic encoder. Let's convert this deterministic encoder to a stochastic encoder that outputs a Gaussian z with a fixed variance $\epsilon$ and the mean being the same as its corresponding deterministic version. The theory tells us that, by minimizing the first W-distance over all sampled z's from this stochastic encoder and the standard cross-entropy loss, we will automatically minimize the second W-distance and preserve the global label frequency in the dataset, even though these z's are only $\epsilon$-close to the deterministic encoding features of training data. Moreover, we carefully read and analyzed the unpublished arXiv paper "Adversarial Examples Are not Bugs, They Are Features". We find that minimizing the W-distance somehow helps to make sure that the encoder q(z|x) will identify some robust features instead of non-robust features, because our proposed regularization constrains the $\epsilon$-ball around each q(z|x) to contribute to preserving the global label distribution in the dataset, even with x integrated out. Replacing W-distance with KL divergence loses all these nice properties.  We use a simple but nice-shaped Gaussian prior for p(z) for W-distance minimization to constrain the global shape of the latent embeddings, while permitting high freedom for the shapes of individual class distributions of latent embeddings. We want the classifier to decide the optimal class-specific distributions of latent embeddings. In addition, it is interesting to explore how to set $\epsilon$-ball to make sure the stochastic encoder to best align the latent embedding z to human-perceived robust features, which will be left as future work.
> >
> > We thank the reviewer for pointing us to some interesting arXiv papers that we did not know. We will add the mentioned papers to the reference.  Still, we feel that, our paper is a purely empirical paper proposing a novel optimal transport regularization with some impressive and surprising empirical results and visualizations.  The theory part is important but not essential to the contributions of our empirical method paper. We can move the theoretical analysis part along with all these discussions to the appendix, or we can write another theory paper focusing on the theoretical analysis using a simplified model as done by the arXiv paper mentioned by the reviewer, which will be future work.
> >
> > We hope that the reviewer can re-evaluate our contributions based on the merits of the method and the results. We believe that our surprising results deserve to be shared with the ICLR research community, which might inspire other researchers to perform deeper theoretical analyses than what we did above and plan to do.

---

> > ### Comment · AnonReviewer1 · 2019-11-15
> > **Response**
> >
> > I thank the authors for their response.
> > Overall, I still believe that this paper requires additional work to improve and clarify the presentation and experimental evaluation.
> >
> > Concerning the evaluation, the fact that the adversarial examples with large epsilon can still be recognized by humans doesn't really address the issue I raised. It is very possible to create adversarial examples with epsilon=0.4 that fool humans. Your attacks maybe just don't find them.
> > In any case, the thing that you should be verifying is that your model's accuracy monotonically goes to 0 as epsilon gets closer to 0.5

---

> ### Author Response · Authors · 2019-11-07
> **Response to Official Blind Review #1 (Part 1)**
>
> Thank you for taking time to read the paper and provide reviews!
>
> As for the main concerns:
>
> (1)(2) "Why compressing the data would remove such features". "Why constraining the compressed data to be close to a Gaussian would have a positive effect on robustness"
>
>       Based on the paper "adversarial vulnerability of neural networks increases with input dimension", the vulnerability of DNNs increases with the input dimension. Experiments also imply that it is harder to defend against adversarial examples on datasets with larger input dimensions. Therefore, projecting data into a low-dimensional manifold can help improve the robustness of DNNs.
>
>       However, we found that just naively reducing the dimension in the embedding space does not result in improvement in classifier robustness (see comparison of E-CLA and ER-CLA in 4.2). The reason might be that, although we have reduced the dimension a lot (4 for MNIST, 16 for CIFAR10, 20 for TinyImagenet), the regularization is not enough and the attacker can still find adversarial examples through the gradient. The "pathological" mappings mean the Encoder obtained by naively reducing the embedding dimension without any constraint. The word "pathological" is not properly used and "non-robust" might be a better word to describe the mappings here. Sorry for the confusion caused. This sentence in the introduction means that arbitrarily projecting the input to a low-dimensional space does not necessarily lead to improvement in classifier robustness.
>
>       This is why we bring in the embedding regularization, which further imposes restrictions on the embedding space. Intuitively, we hope that this restriction can help remove the adversarial noise. Since the training process will push the output of the Encoder towards the prior distribution, in the test phase, the embeddings of adversarial examples will also be pushed towards the prior distribution. This process may help reduce the effect of adversarial perturbations. The experiments showed that the regularization process does help. We visualized the embeddings trained by the two different processes with and without regularization in Appendix. Based on the plots, the embeddings produced by the encoder with regularization is well separated in different classes and very compact, compared with the embeddings produced by the encoder without regularization.
>
>      A further question might be "since you found reducing dimension does not help, why do you still choose to reduce embedding dimension? why not just only regularize the embeddings?" We tried different embedding dimensions in our experiments and the results are shown in the Appendix. The results of very large dimensions, such as 256, 512, are not listed since the results are bad compared to small dimensions. The reason might be that it is easier for the framework to converge on smaller dimensions.

---

### Official Review · AnonReviewer3 · 2019-10-22
**Official Blind Review #3**

**Rating:** 6

**Review:**

This paper proposes a new regularization technique called Embedding Regularization to improve the adversarial robustness.  The idea is to use generative adversarial networks (GAN) to perform inference on the latent space by matching the aggregated posterior of the hidden space vector with a prior distribution. The proposed strategy could be combined with adversarial training to achieve state-of-the-art adversarial accuracy on several benchmark datasets.

Overall, I find the idea interesting and the experimental results promising. The following are my detailed comments.

a. About the algorithm
The idea of incorporating a GAN based model to regularize the representation learning is not completely novel. In [1], a similar approach has been considered for unsupervised learning (auto-encoders).  Besides the omission of reference, a few points about the algorithm need to be clarified :

a.1 What is the motivation of using the different of discriminator as loss in line 7 of Algorithm 1?
In the standard GAN literature, the discriminator loss is usually in the form log(D(z_{true})) + log (1 - D(z_{generated})). Is there any intuition to prefer the current formulation comparing to the this standard formulation?

a.2 Why do we separate the training of classification loss and regularized loss (line 8 and line 9)?
From the optimization perspective, there is no difficulty to optimize 8+9 together, which corresponds better to the loss described in equation (2). (Just fix the discriminator and train the encoder and classifier jointly) Is there any reason to prefer the current alternative training rather than joint training?

a.3 How large are the discriminator loss versus generator loss?
In the standard GAN training, it is crucial to balance the discriminator loss and the generator loss. A plot comparing the different loss in line 7, line 8  and line 9 will be helpful to understand the role of discriminator in the framework. In particular,  it would be interesting to see whether there is a difference when combining with/without adversarial training.

b. About the experiments
b.1 Why Gaussian distribution is a good prior distribution?
Intuitively, we would expect the latent distribution to be well clustered in several class which is clearly not the property of a Gaussian distribution. It is very curious to me why imposing a Gaussian distribution could improve the robustness. Have you tried to use clustered distribution as a prior? (examples could be find in [1]) If not, it would be interesting to try and compare the results.

b.2 On CIFAR10, why the non-robust training outperform the robust training?
It is curious to see that on CIFAR10, ER-classifier- outperforms ER-Classifier with large epsilon. This is surprising since the min-max robust training explicitly minimize the robust loss.

b.3 Have you tried to vary the regularization parameter lambda?
It would be interesting to see how the performance changes when varying the regularization parameter lambda. By the way, how is lambda determined in the current experiments?

Minor comments:
b.4 There is a stronger version of Defense-GAN introduced in [2]
b.5 The study on the dimension of embedding space is interesting, is the non robust training giving similar results?

[1] Adversarial Autoencoders, Makhzani et al. 2015
[2] The Robust Manifold Defense: Adversarial Training using Generative Models, Ilyas et al, 2017

**Experience Assessment:**

I have read many papers in this area.

**Review Assessment: Checking Correctness Of Derivations And Theory:**

I assessed the sensibility of the derivations and theory.

**Review Assessment: Checking Correctness Of Experiments:**

I assessed the sensibility of the experiments.

**Review Assessment: Thoroughness In Paper Reading:**

I read the paper thoroughly.

---

> ### Author Response · Authors · 2019-11-05
> **Response to Official Blind Review #3**
>
> Thanks for taking time to read the paper and the detailed comments!
>
> a. About the algorithm and novelty:
>
> We'll add "Adversarial Autoencoders, Makhzani et al. 2015" in the reference.
>
> Adding a Wasserstein distance regularization in a bottleneck layer of a deep supervised classifier is original and has profound impact on defending against adversarial examples, which has rigorous theoretical support. Please see Part 2 and 4 of our response to Reviewer 1 for details.
>
> As far as we know, our method is original in the following aspects: (1) Although incorporating a GAN based model to regularize the representation learning is widely used for unsupervised learning, our method is the FIRST that applies a Wasserstein distance regularization for the low-dimensional embedding layer in a supervised setting without considering any reconstruction loss; (2) This proposed framework is proved to minimize the optimal transport cost between marginal label distribution in training data and the output distribution of the framework, in which the classifier can be viewed as a generator for generating labels from latent embeddings that preserve global label frequency in the dataset while minimizing cross-entropy loss at the same time;  (3) Our method is also the FIRST that establishes the connection between this Wasserstein distance regularization and robustness of deep neural networks for defending against adversarial examples.
>
> For these reasons, we believe that our method is original and the simple regularization is profound in a supervised learning setting.
>
> a.1 What is the motivation of using the different of discriminator as loss in line 7 of Algorithm 1?
>
>     Yes, in the standard GAN literature, the discriminator loss is in the form log(D(z_{true})) + log (1 - D(z_{generated})).
>
>     We used E(D(z_{true}))- E(D(z_{generated})) (see Algorithm 1), which is Wasserstein distance in Wasserstein GAN (WGAN). In fact, we have tried both objectives, and found that Wasserstein distance minimizing an optimal transport cost yields much better results.
>
> a.2 Why do we separate the training of classification loss and regularized loss (line 8 and line 9)?
>
>      The regularized loss will not be applied to update the Classifier part so we separated the training of the two losses. Thanks for pointing this out! We'll make it more clear.
>
> a.3 How large are the discriminator loss versus generator loss?
>
>      We didn't show this but the two losses are on the same scale when the training process convergences. Both you and reviewer 2 mentioned this. We'll add a plot of the two losses in the Appendix to visualize the training process.
>
> b. About the experiments
>
> b.1 Why Gaussian distribution is a good prior distribution?
>
>      Yes, intuitively we would expect the latent distribution to be well clustered in several classes. In fact, we have tried GMM and expected it to outperform standard Gaussian. However, choosing GMM as prior, we'll have to specify too many hyper-parameters and the experiment results were not as good as expected.
>
>     In fact, in the current framework, the classification loss naturally does the job of separating the embeddings into different classes since the classification loss will be used to update both Encoder and Classifier. Based on the embedding visualization plots in Appendix, we can see that ER-Classifier has latent distribution to be well clustered in several classes but more compact than regular classifier. Thanks for bring this point out!
>
> b.2 On CIFAR10, why the non-robust training outperform the robust training?
>
>   This actually also surprises us. We don't have a clear idea of the reason. We visualized the embedding space of ER-classifier- and ER-Classifier, and found that when combined with min-max optimization, the embedding space of ER-Classifier is not as "good" as ER-Classifier-, where "good" means different classes are well separated and compact. This indicates that min-max optimization somehow has counter effect on embedding regularization.
>
>   However, we conducted black-box attack to evaluate the defense methods (ER-Classifier, Madry's adv, ER-Classifier-). Under black-box setting, ER-Classifier outperforms the other two. This indicates that there might be obfuscated gradient issue with ER-Classifier-, since PGD is based on gradient.
>
> b.3 Have you tried to vary the regularization parameter lambda?
>
>    Yes, we tried to vary the lambda. Currently, lambda is determined by trial with a validation set by splitting the training data. The lambda will influence the robustness and accuracy of the framework. Generally speaking, the large lambda will affect the classification accuracy. While if the lambda is too small, the robustness is affected.
>
> b.4-b.5 Thanks for pointing out the latest version! As for the dimension study, it is based on non-robust training. We didn't test it on robust training setting since robust training costs a lot of time.

---

### Official Review · AnonReviewer2 · 2019-10-27
**Official Blind Review #2**

**Rating:** 3

**Review:**

The paper proposed a adversarial learning framework that tries to align the hidden features of data with simple prior distributions.  A training strategy similar to GAN  was exploited. The proposed framework was argued that it can well deal with the adversarial perturbations. Some experiments were conducted, verifying that the proposed algorithm seems useful and robust.

The main comments are listed as follows:

(1) To the reviewer, the adversarial framework to use a simple prior to align the hidden features seems new. However, in the literatures, there are some adversarial training algorithms which did similar things. For example, C1 tried to force the hidden features of deep learning follow a Gaussian distribution with the smooth regularizer. Though different,  I believe they share some common motivation, the authors may want to discuss and even compare this reference.
Similarly, in the literature, the so-called Center Loss will also basically force the means of different data are as further as possible, this is also relevant to the paper.

(2) The authors have reported a series of experiments, which is great. However, they only evaluated their model's robustness in case of PGD attack. This is very different from many adversarial literature which normally would discuss various attack such as l_2 attack, FSGM, and even black-box attack.  The evaluations may be more convincing  if more attacks can be tested on the proposed method.

(3) Further to (2), it is noted that the paper simply compared their approach's robustness  with Madry's adv, it would be more convincing to compare the other recently adversarial training algorithms like VAT (C2).

(4) It is good that the paper proposed a different way in dealing with adversarial examples. In comparison, the current work studying adversarial examples is based on the robust framework of minimax trying to impose the worst-case perturbation. It would be interesting to see if the proposed work can be also further applied in the minimax  framework and examine if a further robustness can be achieved.

(5) Some visualization may be interesting. For example, a visualization how Q would change as the training continues. This may be used to check the convergence property of the proposed algorithm.

C1:Manifold Adversarial Learning, S. Zhang et al.,  https://arxiv.org/pdf/1807.05832
C2:Virtual Adversarial Training: a Regularization Method for Supervised and Semi-supervised Learning, T. Miyato et al. arXiv preprint arXiv:1704.03976, 2017.


==================

Thanks for the response made by the authors. Thought these response resolved some of my concerns, it is still not very convincing.  On one hand, it seems that the authors did not  comment (4) and (5).  On the other hand, the paper may be still in lack of comparisons and/or discussions with the related work. After PGD, there are a lot of new robust approaches. Overall, I enjoy reading this paper but I would still think the paper could have further room to be  enhanced.

**Experience Assessment:**

I have published one or two papers in this area.

**Review Assessment: Checking Correctness Of Derivations And Theory:**

I assessed the sensibility of the derivations and theory.

**Review Assessment: Checking Correctness Of Experiments:**

I assessed the sensibility of the experiments.

**Review Assessment: Thoroughness In Paper Reading:**

I read the paper at least twice and used my best judgement in assessing the paper.

---

> ### Author Response · Authors · 2019-11-05
> **Response to Official Blind Review #2**
>
> Thank you for taking time to read the paper and provide reviews!
>
> As for the main comments:
>
> Response to Comment (1):
>
> For the novelty and significance of our method, please check our response to Question a of Review #3.
>
> We just read "Manifold Adversarial Learning", which indeed shares related ideas and tries to regularize the latent space with Gaussian Mixture Model and applies KL-divergence to do the optimization.
>
> The difference between "Manifold Adversarial Learning" and "ER-Classifier" is that ER-Classifier does not regularize the embedding space with GMM and we used Wasserstein distance instead of KL-divergence. It definitely worth comparing ER-Classifier with Manifold Adversarial training, but we didn't find the code of it online because this paper has not been officially published in any peer-reviewed conference or journal yet. We'll cite the paper but will only compare with it when the code is available.
>
> Moreover, in ER-Classifier, we don't have a restriction on the prior, but found that standard Gaussian achieves good results.  We use the simple but nice-shaped Gaussian prior for p(z) for Wasserstein distance minimization to constrain the global shape of the latent embeddings, while permitting high freedom for the shapes of individual class distributions of latent embeddings. We want the classifier to decide the optimal class-specific distributions of latent embeddings. In fact we have tried GMM as a prior, but it does not achieve expected results defending against attacks. The reason might be that it is hard to tune hyper-parameters for GMM in our framework or GMM does not work well in our framework.
>
> The advantage of KL-divergence is that people don't have to train a discriminator to minimize the distance. However, we have tried KL-divergence under our framework and found that Wasserstein distance yields better results, which is shown in 4.2 comparing VAE-CLA with ER-CLA. Please check Part  2 and 4 of our response to Review #1 for explaining why Wasserstein distance regularization is better than KL divergence.
>
> Moreover, please note that the Manifold Adversarial training was not designed for defending against adversarial examples but for improving test performance  in the arxiv paper, which is different from the main purpose of our method.
>
> Response to Comment (2):
>
> We have just evaluated the baselines against the black-box attack Nattack and found that ER-Classifier is still better than other methods under this setting. Please see our responses to public comments in this thread. We have added this part into the paper.
>
> We evaluated all the methods against PGD since the two major baselines (Madry's adv, RSE) are mainly evaluated by PGD in their papers.
>
> When comparing with Defense-GAN, we tested two methods against CW attack with L2-norm for fair comparisons. We didn't use FGSM since it is relatively weak compared with PGD and CW.
>
> Response to Comment (3)-(5):
>
> Thanks for the suggesting the VAT paper: C2: Virtual Adversarial Training: a Regularization Method for Supervised and Semi-supervised Learning, T. Miyato et al. https://ieeexplore.ieee.org/stamp/stamp.jsp?tp=&arnumber=8417973   We will cite the paper and discuss how it's related to our method.
>
> However, we carefully read this paper and found that the method was not designed as a defense method against adversarial examples either, which is completely different from our main purpose. Instead, it's proposed for improving generalization performance beating competing methods such as highway networks, DenseNet, and ResNet. It's called Virtual Adversarial Training as opposed to Real Adversarial Training, because it employs virtual labels generated by current predictors to identify search directions that can smooth the output label distribution of classifier. Therefore, it is perfectly suitable for semi-supervised learning like regularized label propagation algorithms for improving generalization performance.
>
> Please note that the final version of this paper was published in August 2019.  Although the final version was only published three months ago, the authors did not compare VAT with any defense method against adversarial attacks, which suggests that the purpose of VAT is indeed completely different from ours. If time permits, we'll add VAT as a baseline in the experiment. It is very likely that VAT will perform much worse than our baseline Madry's adv, because it is not even designed for defending against real (not virtual) adversarial attacks.

---

### Public Comment · ~Anthony_Wittmer1 · 2019-09-29
**evaluation questions**

Thanks for this contribution to the community and I have some questions about the evaluation.

For the CIFAR-10 evaluation in Figure 2, it is hard to believe that ER-Classifier^{-}  outperforms the PGD-adversarial training because ER-Classifier^{-} does not use the adversarial examples during training. In addition, most baselines in this paper have been broken by BPDA[1] or Nattack[2], except adversarial training(Madry et al.). For example, RSE has been broken by Nattack with 100% attack success rate. TRADES[3] is a better baseline for adversarial defense.

For the obfuscated gradients, the evaluation on the black-box attacks is missing in this papaer, so it is hard to know whether the proposed method causes obfuscated gradients[1]. In order to check whether obfuscated gradients has happened, BPDA[1] or Nattack[2] is better choice to evalaute the models.

[1] Obfuscated Gradients Give a False Sense of Security: Circumventing Defenses to Adversarial Examples. ICML 2018
[2] NATTACK: Learning the Distributions of Adversarial Examples for an Improved Black-Box Attack on Deep Neural Networks. ICML 2019
[3] Theoretically Principled Trade-off between Robustness and Accuracy. ICML 2019

---

> ### Author Response · Authors · 2019-10-06
> **RE: evaluation questions**
>
> Thank you for the comment! Sorry for the late reply.
>
> When evaluating the ER-Classifier framework, PGD attacks the whole framework (both the Encoder and Classifier parts). We didn't do anything to mask the gradients. Therefore, PGD equals to BPDA here.
>
> As for the score based attack, we evaluated Madry's adversarial training, ER-Classifier against Nattack on CIFAR10. Nattack is only performed on the first 100 images of CIFAR10 since the attack process takes a long time. For both attack methods, the epsilons are 0.03. The results are listed below:
> Method	                           Madry’s adv  ER-Classifier
> PGD	                                42.3%	         51.3%
> Nattack(100 examples)	40.5%	         47.7%
>
> ER-Classifier still performs better than Madry's adversarial training under score-based attack.

---

> > ### Public Comment · ~Anthony_Wittmer1 · 2019-10-08
> > **Thanks for the additional results.**
> >
> > Thanks for the additional results. Maybe because that the number of testing samples is too small, it seems a bit strange that the black-box attack is stronger than the white-box attack, which is one of the phenomena of obfuscated gradients.
> >
> > From the paper of Nattack, the accuracy of Madry’s adv on Nattack is 52.1%. In addition, the reported accuracy (42.3%)  about  Madry’s adv on PGD attack (epsilon=0.03) is lower than the accuracy (45.8%) on the paper of Madry.
> >
> > Have the authors tried to perform Nattack on ER-Classifier^{-}, which trains without adversarial examples. As mentioned above, I think that ER-Classifier^{-}  may be weaker than Madry’s adv.

---

> > > ### Author Response · Authors · 2019-10-08
> > > **RE: evaluation questions**
> > >
> > > Thank you for the comment! We tried Nattack on ER-Classifier^{-}. The accuracy drops to 34.4%, worse than Madry's adversarial training but better than the baseline model (No defense).
> > >
> > > As for the situation of "the reported accuracy (42.3%)  about  Madry’s adv on PGD attack (epsilon=0.03) is lower than the accuracy (45.8%) on the paper of Madry.", since the difference is not large, it could be a random issue or related to the model structures.
> > >
> > > Thank you again for the comments!

---

> > ### Comment · AnonReviewer1 · 2019-10-17
> > **Percentages are too accurate.**
> >
> > How do you get percentages with decimal precision if you only evaluate on 100 images?

---

> > > ### Author Response · Authors · 2019-10-17
> > > **RE: Percentages are too accurate**
> > >
> > > Sorry for the confusion caused! The originally wrongly classified images are skipped. So 100 images are evaluated but the denominator is not 100. The nattack evaluates defense methods by attack success rate, which equals to number of success attack/number of attacked images (not exactly 100). We reported accuracy equals to (1-success rate) here. Thanks for the comment!

---

> > > > ### Public Comment · ~Anthony_Wittmer1 · 2019-10-18
> > > > **Evaluation metric**
> > > >
> > > > Hi,
> > > >
> > > > For a new attack, it is right to make evaluations with the metric "success rate", so as to remove the effect of some testing images, which have already been misclassified by the target model. In other word, there is no need to copy with the misclassified  clean images.
> > > >
> > > > However, for a new defense, it is not right to make evaluations with the metric "success rate" of the attack, since different models havs different standard accuracies on the clean images. Hence, to align the robust accuracy on PGD,  it is better to reporte accuracy equals to "fail num / evaluation num" for Nattack, since the implement code[1] of Nattack skips the the misclassified  clean images.
> > > >
> > > > [1] https://github.com/Cold-Winter/Nattack

---

> > > > > ### Author Response · Authors · 2019-10-22
> > > > > **RE: Evaluation metric**
> > > > >
> > > > > Thank you for the comment! We re-evaluated all three methods with Nattack accuracy, where accuracy = number of correctly classified/number of attacked images (exactly 100).
> > > > >
> > > > > Method	                                     Madry’s adv  ER-Classifier  ER-Classifier-
> > > > > Nattack(100 examples acc)	38%	                 43%             32%
> > > > >
> > > > > Since the originally wrongly classified images are also taken into account, the accuracies drop for all three methods. But ER-Classifier is better than Madry's adv.

---

### Author Response · Authors · 2019-11-13
**Paper Revision**


Dear reviewers,

Thank you very much for taking time to read our paper and provide detailed reviews! Based on your comments, we have revised the paper to address your concerns:

1. We have added all the papers recommended by the reviewers to our reference and discussed their relations with our method in the related work.

2. We significantly improve the presentation of our method (part 3) to make it easier to follow. The presentation of the theory part is significantly improved and moved to the Appendix in a self-contained way. Again, we emphasize that our paper is an empirical method paper. The  theory is important to explain our motivation and justification of our proposed method, but is not  one of our major contributions. Therefore, we only provide high-level explanations of our theory and justifications of our method using plain descriptions in part 3.

3. High-resolution visualization figures of test data and adversarial images are added to the appendix (please check the embedding visualization section in the appendix for details).

4. The method is not built on either MagNet or DefenseGAN. The comparison with DefenseGAN is not important and only 1% of the whole experiment (also highlighted in blue in the experiment).

5. Comparison to Nattack under Blackbox attack is added.

6. Novelty:  (1) Our method is the first that applies a Wasserstein distance regularization to a bottleneck embedding layer of a deep neural network in a purely supervised setting without considering any reconstruction loss;  (2) Our method is the first that establishes the connection between a Wasserstein distance regularization and robustness of deep neural networks for defending against adversarial examples.

Main changes are marked in blue in the new version. Thanks again for your valuable suggestions and time!

---

### Note · Authors · 2019-12-20
**Submission Withdrawn by the Authors**

I have read and agree with the venue's withdrawal policy on behalf of myself and my co-authors.

---

### Decision · Program_Chairs · 2019-12-19

**Decision:**

Reject

**Comment:**

The paper suggests a new way to defend against adversarial attacks on neural networks. Two of the reviewers were negative, one of them (the most experienced in the subarea) strongly negative. One reviewer is weakly positive. The main two concerns of the reviewers are insufficient comparisons with SOTA and lack of clarity. The authors' response, though detailed, has not convinced the reviewers and has not alleviated their concerns.